# CROM: Continuous Reduced-Order Modeling of PDEs Using Implicit Neural Representations

**Peter Yichen Chen[1,3]**   **Jinxu Xiang[1]**   **Dong Heon Cho[1]**   **Yue Chang[4]**   **G A Pershing[1]**

**Henrique Teles Maia[1]**   **Maurizio M. Chiaramonte[2]**   **Kevin Carlberg[2]**   **Eitan Grinspun[4,1]**

[1]Columbia University   [2]Meta Reality Labs Research   [3]MIT CSAIL   [4]University of Toronto

## ABSTRACT

The long runtime of high-fidelity partial differential equation (PDE) solvers makes them unsuitable for time-critical applications. We propose to accelerate PDE solvers using reduced-order modeling (ROM). Whereas prior ROM approaches reduce the dimensionality of *discretized* vector fields, our *continuous* reduced-order modeling (CROM) approach builds a low-dimensional embedding of the *continuous* vector fields themselves, not their discretization. We represent this reduced manifold using continuously differentiable neural fields, which may train on any and all available numerical solutions of the continuous system, even when they are obtained using diverse methods or discretizations. We validate our approach on an extensive range of PDEs with training data from voxel grids, meshes, and point clouds. Compared to prior discretization-dependent ROM methods, such as linear subspace proper orthogonal decomposition (POD) and nonlinear manifold neural-network-based autoencoders, CROM features higher accuracy, lower memory consumption, dynamically adaptive resolutions, and applicability to any discretization. For equal latent space dimension, CROM exhibits $79\times$ and $49\times$ better accuracy, and $39\times$ and $132\times$ smaller memory footprint, than POD and autoencoder methods, respectively. Experiments demonstrate $109\times$ and $89\times$ wall-clock speedups over unreduced models on CPUs and GPUs, respectively. Videos and codes are available on the project page: `https://crom-pde.github.io`.

## 1 INTRODUCTION

Many scientific and engineering models are posed as partial differential equations (PDEs) of the form

$$\mathcal{F}(\boldsymbol{f}, \boldsymbol{\nabla}\boldsymbol{f}, \boldsymbol{\nabla}^2\boldsymbol{f}, \ldots, \dot{\boldsymbol{f}}, \ddot{\boldsymbol{f}}, \ldots) = \mathbf{0}, \quad \boldsymbol{f}(\boldsymbol{x}, t) : \Omega \times \mathcal{T} \to \mathbb{R}^d, \tag{1}$$

subject to initial and boundary conditions. Here $\boldsymbol{f}$ is a spatiotemporal dependent, multidimensional continuous vector field, such as temperature, velocity, or displacement; $\boldsymbol{\nabla}$ and $\dot{(\cdot)}$ are the spatial and temporal gradients; $\Omega \subset \mathbb{R}^m$ and $\mathcal{T} \subset \mathbb{R}$ are the spatial and temporal domains, respectively.

We may solve for $\boldsymbol{f}$ by discretizing in space, $\boldsymbol{f}(\boldsymbol{x}, t) \approx \boldsymbol{f}_P(\boldsymbol{x}, t) = \sum_{i=1}^{P} \boldsymbol{a}^i(t)N^i(\boldsymbol{x})$, transforming the continuous spatial representation to a $(P \cdot d)$-dimensional vector whose coefficients $\boldsymbol{a}^i(t) : \mathcal{T} \to \mathbb{R}^d$ and the corresponding basis functions $N^i(\boldsymbol{x}) : \Omega \to \mathbb{R}$ (e.g., polynomial basis, fourier basis) approximate the continuous solution. For instance, if $N^i$ is the linear finite element basis, the coefficients $\boldsymbol{a}^i(t) = \boldsymbol{f}(\boldsymbol{x}^i, t)$ are field values at spatial samples $\boldsymbol{x}^i$ (Hughes, 2012).

After introducing temporal samples $\{t_n\}_{n=0}^{T}$, we temporally evolve the solution by solving for $P$ unknowns $\{\boldsymbol{a}^i(t_{n+1})\}$ given the previous state $\{\boldsymbol{a}^i(t_n)\}$. Unfortunately, when $P$ is large, processing and memory costs of these *full-order* solves become intractable. To alleviate this computational burden, prior model reduction techniques (Berkooz et al., 1993; Willcox & Peraire, 2002; Benner et al., 2015) construct a manifold-parameterization function $\boldsymbol{g}_P : \mathbb{R}^r \mapsto \mathbb{R}^{Pd}$, with $r \ll Pd$, such that every low-dimensional latent space vector $\boldsymbol{q}(t) \in \mathbb{R}^r$ maps to a discrete field $\boldsymbol{g}_P(\boldsymbol{q}) \mapsto (\boldsymbol{a}^1, \ldots, \boldsymbol{a}^P)^T$. For instance, for linear finite elements (Barbič & James, 2005), $\boldsymbol{g}_P(\boldsymbol{q}) \mapsto \left(\boldsymbol{f}(\boldsymbol{x}^1, t), \ldots \boldsymbol{f}(\boldsymbol{x}^P, t)\right)^T$,

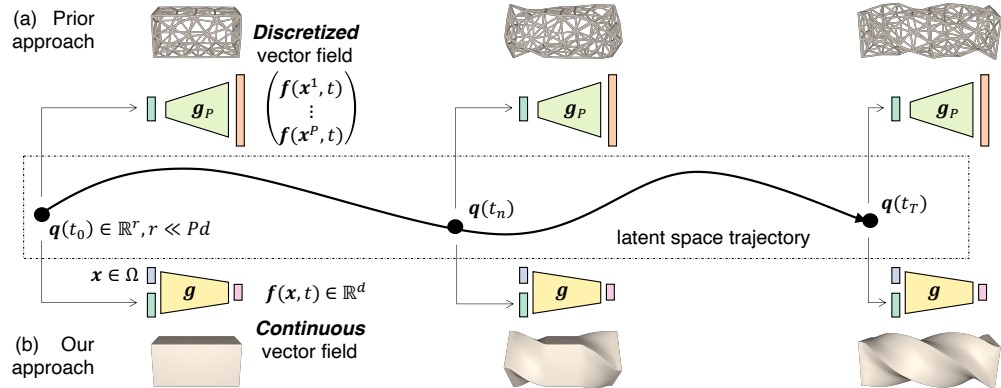

Figure 1: Model reduction solves PDEs via temporal evolution of the low-dimensional latent space vector $q(t)$. (a) Prior work assumes that the low-dimensional representation $g_P$ is built for the *already-discretized* vector field; (b) our approach constructs the manifold-parameterization function $g$ directly for the *continuous* vector field itself. In this case, the vector field $f$ represents the twisting material governed by the elastodynamics equation.

as depicted in Figure 1a. ROM saves computation because it requires evolving only $r \ll Pd$ latent space variables.[1]

Since existing ROM approaches apply to *already-discretized* fields, model training and PDE solving are tied to the dimension and discretization type of the training data, causing key limitations:

**Discretization dependence.** If we alter the training simulation resolution ($P$) or the discretization types (e.g., meshes to point clouds), we must also alter the architecture and numbers of parameters.

**Memory scaling.** Memory footprint grows with discretization resolution $P$.

**Fixed discretization.** We cannot dynamically *adapt* spatial resolution $P$, discretization type, or basis function $N^i$ during latent-space-PDE solves, e.g., dynamic remeshing (Peraire et al., 1987).

Altogether these problems arise because the architecture of $g_P(q)$ is tied to the discretization $(a^1, \ldots, a^P)^T$.

**Introducing a discretization-independent architecture** In an alternative point of departure, we train a manifold-parameterization function $g(x, q) \approx f(x, t)$ to approximate the *continuous* field itself, *not* its discretization (see Figure 1b). Note that the domain and co-domain of $g$ are continuous domains: they do not depend on the choice of discretization(s) used at any stage of the process, i.e., during preparation of training data, nor during latent-space-PDE solving. In this sense, the manifold-parameterization architecture is *discretization independent*. In our implementation, $g$ is embodied as an implicit neural representation (Park et al., 2019; Chen & Zhang, 2019; Mescheder et al., 2019), also known as a neural field, yielding a smooth and analytically-differentiable manifold-parameterization. This representation's memory footprint depends on the complexity of fields produced by the PDE, *not* the discretization *resolution*.

After training, we evolve the latent variables, as governed by the PDE, for previously-unexplored parameters. Unlike approaches that discard the PDE after training, we evaluate the original PDE at a small number of domain points at every time integration step. We validate our approach on classic PDEs with discretized data from voxels, meshes, and point clouds. In comparison to the full-order model, our approach reduces the number of spatial degrees of freedom, memory, and computational cost. In comparison to prior linear and nonlinear discretization-dependent model reduction methods, our method exhibits higher accuracy and consumes less memory. To highlight another benefit of being discretization-agnostic, we demonstrate an elasticity simulation that readily adapts mesh resolution.

## 2 RELATED WORK

**Reduced-Order Modeling for PDEs.** Early works on identifying a low-dimensional latent space focused on linear methods (Berkooz et al., 1993; Holmes et al., 2012), e.g., proper orthogonal decomposition (POD) or principal component analysis (PCA). Recent nonlinear manifolds (Fulton et al., 2019; Lee & Carlberg, 2020), often constructed via autoencoder neural networks, have been

---

[1]The latent space vector is also known as the feature, subspace, or state vector; or the generalized coordinates.

shown to significantly outperform their linear counterparts on slowly decaying Kolmogorov n-width problems (Peherstorfer, 2022). Most of these prior works exclusively focus on building a latent space for the *already-dicretized* vector fields. Chen et al. (2023); Pan et al. (2023) and our work attempt to construct the latent space for the continuous vectors themselves. However, Chen et al. (2023)'s treatment specializes in the material point method (discretization) and elasticity (PDE); our general treatment is both discretization and PDE agnostic. Likewise, Pan et al. (2023) *train* a discretization-agnostic latent space for PDE data; we further *solve* the PDEs in the reduced space via rapid latent space traversal. Specifically, we evolve the latent space in a PDE-constraint manner while Yin et al. (2023) take a data-driven approach. Additional references to ROM are listed in Appendix A.

**Implicit neural representations** use (fully-connected) neural networks to represent arbitrary vector fields. While the Euclidean space spatial coordinates always form part of the input to the network, it is also common to have a latent space vector to complement the rest of the input. Different latent space vectors correspond to different states of the continuous vector field (see Figure 1b), e.g. different geometries (Park et al., 2019; Chen & Zhang, 2019; Mescheder et al., 2019) or different radiance fields (Mildenhall et al., 2020). A key contribution of our work is nonlinearly traversing the latent space of neural representations under an explicit PDE constraint.

**Machine learning (ML) for PDEs.** Physics-informed neural networks (PINNs) (Raissi et al., 2019; Sitzmann et al., 2020b) demonstrate that PDEs can be accurately solved via neural representations. Notably, PINN enables prediction and discovery from incomplete models and incomplete data (Karniadakis et al., 2021). However, the degrees of freedom involved in their approaches are still $Pd$, and the underlying gradient-descent-based solver is often computationally more *expensive* than traditional solvers (see Table 1 by Zehnder et al. (2021)). By contrast, our goal is building a computationally more *efficient* solution that solves for only $r$ degrees of freedom ($r \ll Pd$). In fact, our approach can be viewed as an extension of PINN for model reduction. Setting the latent space vector $\boldsymbol{q}(t)$ in our formulation (see Figure 1b) to the time variable $t$ recovers the exact formulation of PINN. In addition to PINN, Sanchez-Gonzalez et al. (2020) show graph neural network (GNN) architectures are also capable of learning PDEs. Yet, like PINNs, GNNs also do not offer dimension reduction.

## 3 METHOD: OVERVIEW AND MANIFOLD CONSTRUCTION

**Overview** Our goal is to efficiently obtain the solution of Equation (1). We begin by constructing a low-dimensional embedding and the corresponding manifold-parameterization function (see below), after which we solve PDEs by time-integrating the dynamics of the manifold's latent space vector (see Section 4). As we demonstrate in examples from various scientific disciplines (see Section 5), this general method is applicable regardless of the discretization of the training data (e.g., voxel grids, meshes, point clouds) or the discretization deemed most useful for evaluating the gradients (e.g., physical forces) when solving PDEs on the constructed manifold.

**Manifold Construction** As depicted in Figure 1b, we seek a manifold-parameterization function $\boldsymbol{g}(\boldsymbol{x}, \boldsymbol{q})$,

$$\boldsymbol{g}(\boldsymbol{x}, \boldsymbol{q}(t; \boldsymbol{\mu})) \approx \boldsymbol{f}(\boldsymbol{x}, t; \boldsymbol{\mu}), \quad \forall \boldsymbol{x} \in \Omega, \quad \forall t \in \mathcal{T}, \quad \forall \boldsymbol{\mu} \in \mathcal{D}, \tag{2}$$

that well approximates the continuous field $\boldsymbol{f}(\boldsymbol{x}, t; \boldsymbol{\mu})$ throughout the spatiotemporal domain $\Omega \times \mathcal{T}$, and for a workable range of problem parameters $\boldsymbol{\mu} \in \mathcal{D}$. For ease of exposition, we omit the dependencies of $\boldsymbol{q}$ and $\boldsymbol{f}$ on the problem parameters $\boldsymbol{\mu}$. Here $\mathcal{D}$ is an arbitrary parameter space (e.g., material properties, external forces, user settings). For scenarios that do not feature trivial parameterizations, e.g., external force via crowd-sourcing (Barbič & James, 2005), $\mathcal{D}$ can also be implicitly defined.

What distinguishes our approach from typical model reduction is that $\boldsymbol{g}$ takes the position $\boldsymbol{x} \in \Omega$ as an input. Thus, unlike prior approaches that infer only discrete coefficients, our approach infers field values at arbitrary domain positions $\boldsymbol{x}$.

We parameterize $\boldsymbol{g}$ with a neural network $\boldsymbol{g}_{\theta_g}$ whose weights $\theta_g$ satisfy the minimization problem

$$\min_{\theta_g} \sum_{i=1}^{P} \sum_{n=0}^{T} \sum_{\boldsymbol{\mu} \in \mathcal{D}_{\text{train}}} \|\boldsymbol{g}_{\theta_g}(\boldsymbol{x}^i, \boldsymbol{q}(t_n)) - \boldsymbol{f}(\boldsymbol{x}^i, t_n)\|_2^2, \tag{3}$$

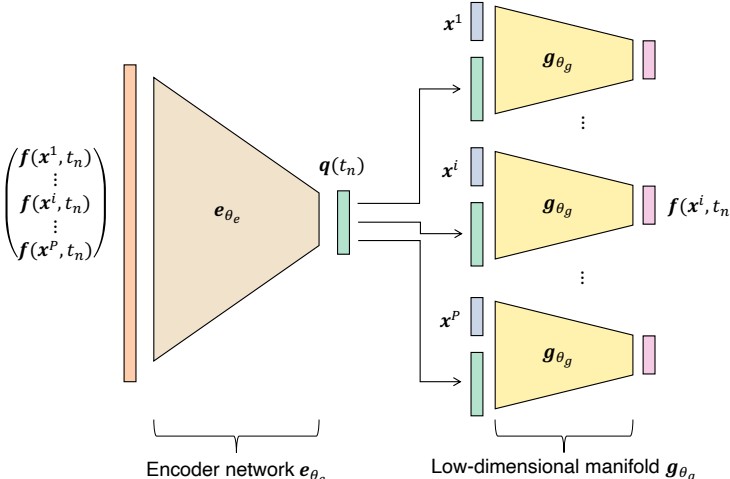

Figure 2: Constructing the manifold-parameterization function as a neural network trained via supervised learning. We pass each snapshot (time step) from the training dataset into an encoder to obtain a latent space vector $q$. We then concatenate $q$ with the spatial coordinates and pass that into the manifold-parameterization function with the goal of reconstructing $f$ for each individual spatial sample. The same $q$ is shared among all spatial samples in this time step.

where $\mathcal{D}_{\text{train}} \subset \mathcal{D}$ is the training set, and $q(t_n)$ is the latent space vector shared among all spatial samples. This objective aims to reproduce all the field values present in the training data, generated via full-order PDE solutions. Notably, our approach imposes no limit on the discretization strategy of the PDE solver. For instance, this framework is applicable to training data from both finite difference methods and finite element methods as well as both voxel grids and meshes.

There are commonly two approaches to define the latent space vector $q$: the auto-decoder approach (Park et al., 2019) that trains $q$ along with $g_{\theta_g}$ and the encoder approach (Chen & Zhang, 2019; Mescheder et al., 2019) that trains a separate network to output $q$. While both approaches work for our application, we adopt the latter.

The encoder network $e_{\theta_e}$ with weights $\theta_e$ takes an input vector constructed by concatenating all the discrete degrees of freedom from the training data and outputs a latent space vector (see Figure 2):

$$e_{\theta_e}(\overrightarrow{f}(t)) = q(t), \quad \text{where } \overrightarrow{f}(t) = \big(f(x^1, t), \dots f(x^i, t), \dots f(x^P, t)\big)^T.$$

We emphasize that this discretization-dependent encoder (Xie et al., 2021) is a tool for training the smoothly varying latent space and for determining the initial latent space vector from the discretized training data. Otherwise, the implicit neural representation $g_{\theta_g}$ remains a discretization-agnostic architecture.

Adding the encoder, Equation (3) now becomes

$$\min_{\theta_g, \theta_e} \sum_{i=1}^{P} \sum_{n=0}^{T} \sum_{\mu \in \mathcal{D}_{\text{train}}} \|g_{\theta_g}(x^i, e_{\theta_e}(\overrightarrow{f}(t_n))) - f(x^i, t_n)\|_2^2. \tag{4}$$

Figure 2 illustrates the training pipeline. Please refer to Appendix D for network and training details and Appendix F for hyperparameter selection.

## 4 METHOD: LATENT SPACE DYNAMICS

After the manifold is constructed, we compute the low-dimensional latent space dynamics ($q_n \mapsto q_{n+1}$) in three steps (see Figure 3 and appendix Q): (1) network inference, (2) PDE time-stepping, and (3) network inversion. The neural network strictly serves as a *kinematic spatial* representation in the maps from and to the latent space (steps 1 and 3, respectively). The time integration (step 2) itself uses the original PDE, not a neural network approximation thereof. The ROM community has demonstrated that these three steps can yield strong long-time stability even on stiff and chaotic dynamical systems (Carlberg et al., 2013; 2017). Appendices O and P demonstrate smooth latent space trajectories and empirical stability analysis using this approach.

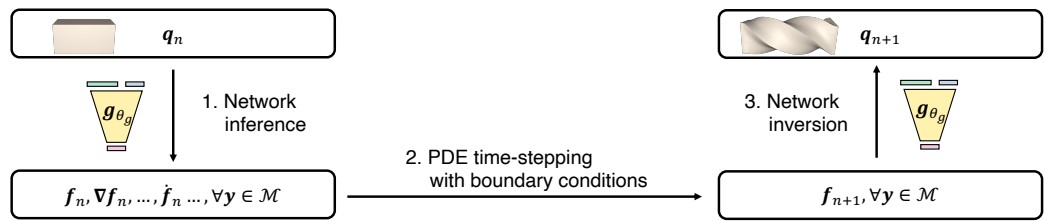

Figure 3: Latent space dynamics: temporally evolve from one latent space vector to another, governed by the PDE. The entire pipeline only involves degrees of freedom from a small spatial subset $\mathcal{M}$, where $|\mathcal{M}| \ll P$.

Commonly shared among all three steps are "integration samples", a finite set of spatial domain points $\mathcal{M} := \{\boldsymbol{y}^j \in \Omega \mid 1 \leq j \leq |\mathcal{M}|\}$ chosen at the user's discretion (see Section 4.4). These samples need not coincide with the previously mentioned full-order finite element discretization samples $\{\boldsymbol{x}^i\}_{i=1}^{P}$.

## 4.1 STEP 1: NETWORK INFERENCE

We first aim to gather all the full-space spatiotemporal information ($\forall \boldsymbol{y} \in \mathcal{M}$) necessary for PDE time integration. The function value $\boldsymbol{f}$ itself can be evaluated via inferencing of the neural network $\boldsymbol{f}(\boldsymbol{y}, t_n) = \boldsymbol{g}_{\theta_g}(\boldsymbol{y}, \boldsymbol{q}_n)$. The spatial and temporal gradients are computed either by differentiating the network, $\boldsymbol{\nabla} \boldsymbol{f}(\boldsymbol{y}, t_n) = \boldsymbol{\nabla}_{\boldsymbol{y}} \boldsymbol{g}_{\theta_g}$ and $\dot{\boldsymbol{f}}(\boldsymbol{y}, t_n) = \frac{\partial \boldsymbol{g}_{\theta_g}}{\partial \boldsymbol{q}} \dot{\boldsymbol{q}}_n$, respectively, or by numerical approximation. Higher-order gradients may be generalized in a similar manner. Further details on gradient computation are listed in Appendix E.

## 4.2 STEP 2: PDE TIME-STEPPING

We now evolve time from $t_n$ to $t_{n+1}$. Unlike end-to-end learning-based latent space dynamics methods (Lusch et al., 2018), we evaluate time derivatives using the exact PDE (1), *not* a learned surrogate. At each integration point $\boldsymbol{y}$, we evaluate the temporal derivative by solving the PDE (1) for $\dot{\boldsymbol{f}}_{n+1}(\boldsymbol{y})$:

$$\mathcal{F}(\boldsymbol{f}_n, \boldsymbol{\nabla} \boldsymbol{f}_n, \ldots, \dot{\boldsymbol{f}}_{n+1}, \ldots) = \boldsymbol{0} \ . \tag{5}$$

We evolve the configuration to time $t_{n+1} = t_n + \Delta t$ using the chosen explicit time integration method $\mathcal{I}_{\mathcal{F}}$ subject to given boundary conditions, e.g., Runge-Kutta methods (Dormand & Prince, 1980):

$$\boldsymbol{f}_{n+1} = \mathcal{I}_{\mathcal{F}}(\Delta t, \boldsymbol{f}_n, \dot{\boldsymbol{f}}_{n+1}, \ldots) \quad \forall \boldsymbol{y} \in \mathcal{M} \ . \tag{6}$$

While this work focuses on explicit time integration, we can also extend the framework for implicit time integration (Carlberg et al., 2017).

## 4.3 STEP 3: NETWORK INVERSION

We project back onto the reduced manifold by finding the corresponding input $\boldsymbol{q}_{n+1}$ that best matches the evolved configuration $\boldsymbol{f}_{n+1}$ in a least-squares sense: (Quarteroni et al., 2014)

$$\min_{\boldsymbol{q}_{n+1} \in \mathbb{R}^r} \sum_{\boldsymbol{y} \in \mathcal{M}} \|\boldsymbol{g}_{\theta_g}(\boldsymbol{y}, \boldsymbol{q}_{n+1}) - \boldsymbol{f}(\boldsymbol{y}, t_{n+1})\|_2^2 \ . \tag{7}$$

The objective is similar to the training loss found in Equation (3), but with two dimensions significantly reduced: the dimension of the unknown $\boldsymbol{q}_{n+1}$, and the summation bound $|\mathcal{M}|$. Consequently, instead of using a stochastic gradient descent type of method, such as the auto-decoder scheme by Park et al. (2019), we achieve rapid inversion using the Gauss-Newton algorithm (Nocedal & Wright, 2006) with conditionally quadratic convergence. Further details are listed in Appendix C.

## 4.4 SPATIAL SAMPLE REDUCTION

A necessary condition for the well-posedness of the least squares formulation from Equation (7) is $r \leq d|\mathcal{M}|$. Since the manifold-parameterization function construction guarantees that $r \ll Pd$, we

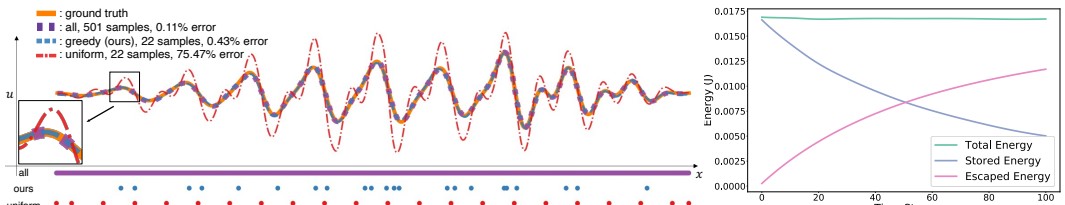

Figure 4: Thermodynamics. Left: integration samples ($\mathcal{M}$). Greedily selecting the samples (blue) allows us to use significantly fewer degrees of freedom than the full-order simulation (purple) while getting a much higher accuracy than naive uniform sampling (red). The depicted field is temperature governed by the heat equation after 100 time steps. Right: as dictated by the PDE, the reduced system is conservative: total energy (stored thermal energy plus cumulative flux at boundary) is conserved.

choose $\frac{r}{d} \leq |\mathcal{M}| \ll P$. To obtain the next-time step $\boldsymbol{f}(\boldsymbol{y}, t_{n+1}), \forall \boldsymbol{y} \in \mathcal{M}$ necessary for the least squares solves, we only require PDE updates (Section 4.2) and spatiotemporal data (Section 4.1) at these $|\mathcal{M}|$ samples. As such, the entire latent space dynamics framework (Figure 3) requires only $|\mathcal{M}|$ samples, compared to the full-order solver's $P$ samples. Hyper-reduction approaches like this have captured a wide range of real-world scenarios, including massive elasticity deformations (Fulton et al., 2019) and large turbulent flows (Grimberg et al., 2021). Unlike our discretization-independent approach, prior methods only support hyper-reduction samples that coincide with the full-order discretization.

A naive selection of the integration samples can lead to inaccurate latent space dynamics, even if $|\mathcal{M}| \geq \frac{r}{d}$; refer to Figure 4 for the failure case of uniform sampling. As noted in the hyper-reduction literature, stochastic sampling can eliminate such errors (Carlberg, 2011). To better *control* hyper-reduction error, we draw inspirations from the cubature approach by An et al. (2008) and propose a greedy algorithm that augments the sample set to meet a target residual (see Appendix B). This strategy iteratively adds spatial samples to the sampling set until the target residual is met. In this case, the user decides the target accuracy, not the sample set size $|\mathcal{M}|$. Alternatively, the user can terminate the algorithm early once it reaches a chosen sample set size $|\mathcal{M}|$. Results of our sampling approach are shown in Figures 4, 9 and 20.

## 5 EXPERIMENTS

We analyze the proposed framework on classic PDEs, with training data produced using a variety of discretizations (voxel grids, meshes, and point clouds). Unless otherwise noted, for each PDE, we delineate a testing set where $\mathcal{D}_{\text{test}} \subset \mathcal{D}$ with $\mathcal{D}_{\text{train}} \cap \mathcal{D}_{\text{test}} = \emptyset$. We construct the manifold (Section 3) with data from $\mathcal{D}_{\text{train}}$, and then validate the latent space dynamics (Section 4) on $\mathcal{D}_{\text{test}}$. We compare our approach with prior discretization-dependent ROM methods, including POD (Berkooz et al., 1993; Holmes et al., 2012) and neural-network-based autoencoder approaches (Fulton et al., 2019; Lee & Carlberg, 2020; Shen et al., 2021), under the same latent space dimensions. Additional implementation and reproducibility details are listed from Appendix G to Appendix L. Experiment statistics are summarized in Table 1. The temporal evolutions of the PDEs are best illustrated via the **supplementary video**.

**Thermodynamics**, $\frac{\partial u}{\partial t} - \nu(x)\frac{\partial^2 u}{\partial x^2} = 0$ . Temperature $u$ is governed by a one-dimensional heat equation. $\nu$ describes the spatially-varying diffusion speed. Figure 4 displays our approach's ability to use very few integration samples and to capture conservation of energy.

**Image Processing**, $\frac{\partial u}{\partial t} - \nu(\boldsymbol{x})\boldsymbol{\nabla}^2 u = 0$ . We model image blurring with the 2D diffusion equation (Perona & Malik, 1990). Figure 5 shows that under the same latent space dimension ($r = 3$), our method is more accurate than POD (Berkooz et al., 1993; Holmes et al., 2012), both visually and quantitatively. With its architecture independent of pixel count, CROM uses an order of magnitude less memory than POD (see Figure 5c).

**Transport dominated systems**. Next, we examine two transport-dominated slowly decaying Kolmogorov n-widths problems, the Advection Equation and Burgers' Equation, where classic model reduction techniques often struggle (Peherstorfer, 2022):

**Advection Equation**, $\frac{\partial u}{\partial t} + (a \cdot \boldsymbol{\nabla})u = 0$ . Here $u$ is the advected quantity and $a$ is the advection velocity. Figure 6a depicts CROM's favorable trajectory tracking relative to both POD and the convolutional autoencoder (CAE) of Lee & Carlberg (2020), for equal latent space dimension (also see Figure 21).

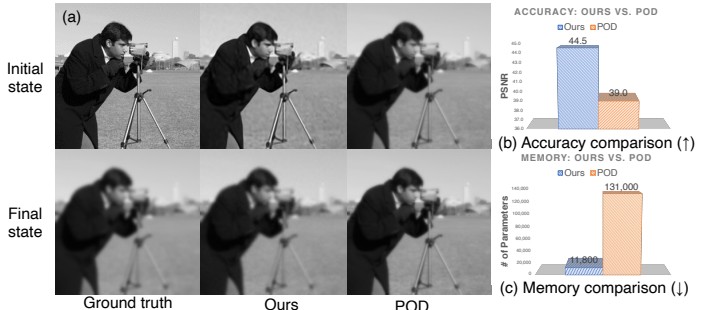

Figure 5: Image processing: comparison with POD. Ground truth solution uses $P = 65,536$ pixels. (a) Visually, our approach better captures the sharp initial state and the smoothed finial state than POD. (b) Quantitatively, CROM obtains a higher PSNR than POD ($\uparrow$ the higher the better). (c) Our approach also uses an order-of-magnitude less memory than POD ($\downarrow$ the lower the better). Both POD and our approach use the same latent space dimension ($r = 3$).

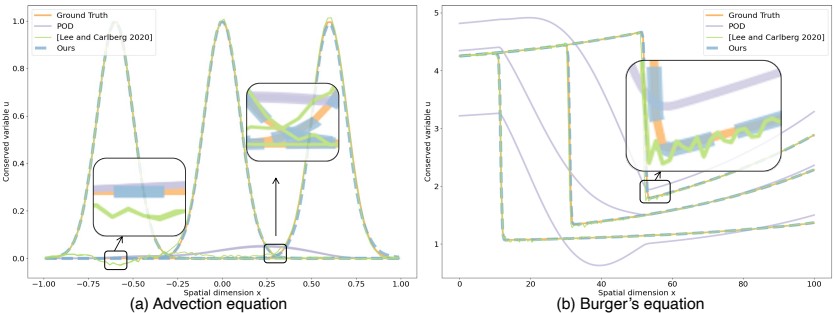

Figure 6: Breaking the Kolmogorov barrier: CROM outperforms POD and convolutional autoencoders (Lee & Carlberg, 2020) in tracking both the Advection and Burgers' trajectories. For both cases, we use the intrinsic solution-manifold dimension (Lee & Carlberg, 2020), the lower bound of $r$, as the latent space dimension $r$. To isolate the source of error, no hyper-reduction method is applied.

**Burgers' Equation**, $\frac{\partial w}{\partial t} + \frac{\partial 0.5w^2}{\partial x} = 0.02e^{\mu_D x}$ . Figure 6b shows that CROM more accurately captures the nonlinear dynamics than both POD and CAE (also see Figure 22). Additionally, CROM uses $12\times$ less memory than CAE (see Figure 27). These accuracy and memory advantages are consistent with the implicit neural representation literature (Chen & Zhang, 2019).

Whereas CAE is applicable only to voxel grid data, CROM is applicable to data from *any discretization*, as highlighted by the tetrahedral mesh and point cloud data examples (see Figure 9).

**Fluid Dynamics**, $\frac{\partial \boldsymbol{u}}{\partial t} + (\boldsymbol{u} \cdot \boldsymbol{\nabla})\boldsymbol{u} = -\boldsymbol{\nabla}p + \nu\boldsymbol{\nabla}^2\boldsymbol{u} + \boldsymbol{f}_{ext}, \boldsymbol{\nabla} \cdot \boldsymbol{u} = 0$. CROM captures the challenging Karman vortex street (see Figure 7), governed by the incompressible Navier-Stokes equations. It even generalizes well beyond the training temporal range.

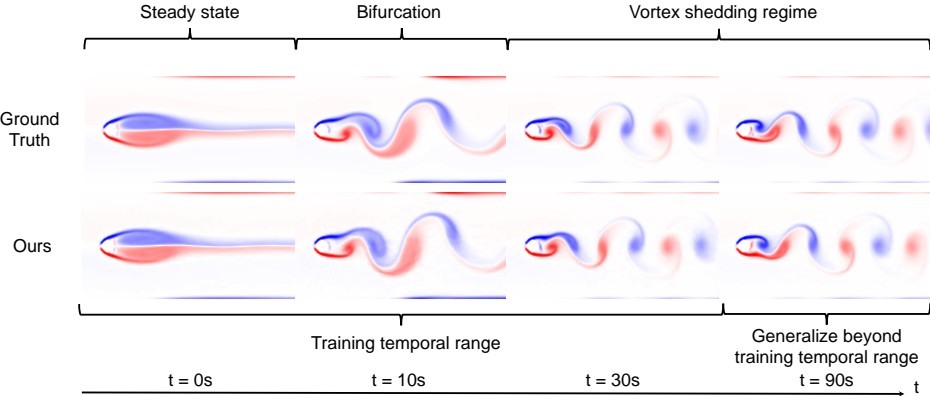

Figure 7: The Karman vortex street. Our latent space dynamics solver accurately captures the flow both before and after the bifurcation: the steady state and the periodic vortex shedding regime. The visualized quantity is the the curl of the velocity field. Our approach even generalize well beyond the training temporal range, thanks to the kinematics-approximation-only nature of CROM.

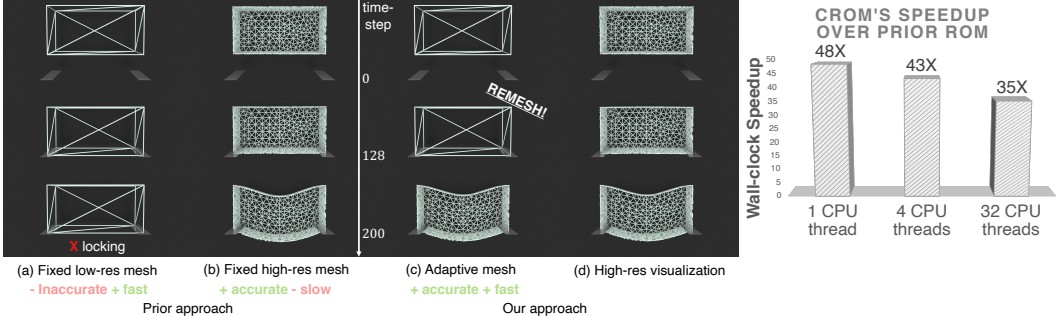

Figure 8: Adaptive discretization in solid mechanics. A falling deformable body impacts two static objects. (a and b) Prior approaches (Barbič & James, 2005; Fulton et al., 2019; Shen et al., 2021) only support reduced-order dynamics with fixed discretizations, forcing a tough choice between accuracy and computational cost; (c) our approach allows dynamic mesh adaptivity to balance accuracy and cost as problem difficulty varies over time. (d) Our resolution-independent encoding of the deformation map allows us to visualize the field using any method (e.g., a high resolution mesh) without regard to the computational mesh. CROM's speedup over prior discretization-dependent ROM methods is observed for 1–32 threads, and emphasized in the case of limited computational resources. To ensure representativeness of prior ROM architectures, we report timing strictly for the PDE-time-stepping stage (Step 2), which is shared among all prior ROM architectures.

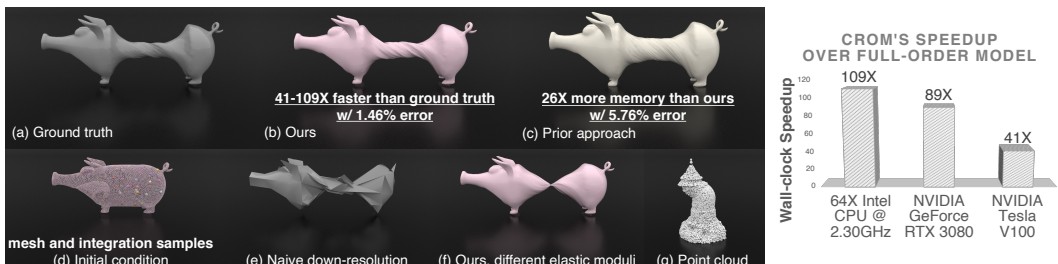

Figure 9: Solid mechanics (a) The ground truth is generated via the full-order PDE solver. (b) Our approach is 41-109× faster than the ground truth while capturing detailed shearing and volume-preserving behaviors (1.46% error). (c) Prior approach (Barbič, 2012) of the same latent space dimension consumes 26× more memory while suffering from volume-gain artifacts (5.76% error). (d) These simulations adopt a tetrahedral discretization (d, pink mesh, $P = 66,608$). Instead of using the expensive high-resolution mesh, our approach computes dynamics using very few integration samples (d, colorful spheres, $|\mathcal{M}| = 40$). (e) Naive down-resolution of the ground truth simulation yields a similar runtime but leads to significantly worse quality. (f) After training, our model can capture a wide range of material properties. (g) The same manifold-parameterization function architecture (different network weights) can also be used for model-reducing point-cloud based simulation (reproduced from the work by (Chen et al., 2023)). The speedup plot on the right demonstrates the effectiveness of our approach on diverse computing platforms. Disclaimer: the authors do not support animal cruelty.

**Solid Mechanics**, $\rho_0 \ddot{\phi} = \nabla \cdot P(\nabla \phi) + \rho_0 B$. We solve the second-order elastodynamics equation for the deformation map $\phi$ of soft bodies, where $\rho_0$ is the initial density, $P$ is the first Piola–Kirchhoff stress, and $B$ is the body force density (see Figures 8 to 10 and 25).

**Adaptive resolution.** Prior ROM methods (see Figure 1a) fix the discretization at onset, precluding the dynamic adaptivity approaches that benefit problems with evolving complexity. When modeling a falling block (see Figure 8), prior ROM methods must begin with a high-resolution mesh, despite that the high-resolution is required only during contact (Figure 8a and b). Our discretization-independent representation allows the time integrator to freely adapt the spatial discretization throughout the dynamic evolution. For instance, we can employ a coarse mesh ($P = 8$ vertices) to economize computation during the rigid falling phase (Figure 8c), and a high-resolution mesh ($P = 2,065$) to capture details during contact (Figure 8c), yielding 35×–48× wall-clock speedups over fixed discretization ROM methods. To ensure generalizability of the comparison, the timing is done strictly for the PDE time-stepping stage (Step 2). Prior ROM architectures (i.e, both POD and autoencoder approaches) share the same PDE time-stepping routine.

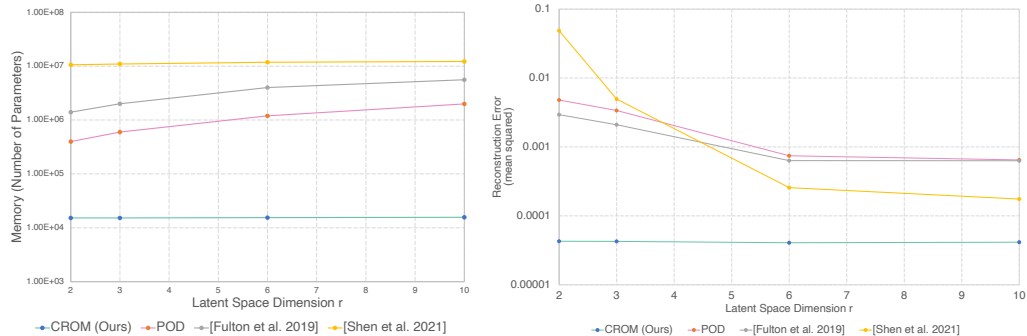

Figure 10: Our approach uses significantly less memory and simultaneously yields higher accuracy than POD and the autoencoder-based ROM methods of Fulton et al. (2019) and Shen et al. (2021) for the solid mechanics experiment depicted in Figure 9. For example, with $r = 3$, our approach uses $39\times$ and $132\times$ less memory, while simultaneously offering $79\times$ and $49\times$ more accuracy, compared to POD and the autoencoder approach of Fulton et al. (2019), respectively.

**Memory footprint.** Most prior ROM approaches are discretization-dependent. As discretization resolution ($P$) increases, so does memory consumption for $\boldsymbol{g}$. By contrast, our discretization-independent architecture does not "see" the parameter $P$ and in practice memory footprint does not scale with $P$. Rather, as with all neural fields (Xie et al., 2021), memory grows with the intrinsic complexity of the training data.

Figure 10 compares memory footprint and reconstruction error for CROM, POD (Barbič, 2012), and neural-network-based autoencoder approaches (Fulton et al., 2019; Shen et al., 2021), as a function of latent space dimension; Figure 9 depicts the corresponding large deformation simulations. Figure 25 compares CROM's memory and accuracy on another large deformation example. For these examples, CROM offers simultaneously lower memory consumption *and* reconstruction error. Furthermore, due to hyper-reduction (from $P = 66,608$ to $|\mathcal{M}| = 40$), CROM also obtains significant wall-clock speedups on CPUs, consumer GPUs, and data center GPUs over the unreduced model (see Figure 9).

**Different discretizations, same architecture.** Because CROM is not constructed for a fixed discretization, we can adopt an identical manifold-parameterization function architecture to model-reduce simulations of both tetrahedral meshes and point clouds (see Figure 9g). Moreover, even though these examples (Figures 8 and 9) leverage completely different meshes, they use the same network architecture. This opens the door for future work on transfer learning (Weiss et al., 2016) and weight sharing among various discretizations and PDEs.

Additional comparisons between CROM, previous ROM, and the full-order models are discussed in Appendices M and N.

## 6 DISCUSSION AND CONCLUSION

CROM is a model reduction framework for PDEs featuring a discretization-independent architecture. CROM disentangles the low-dimensional embedding from the discretization by reformulating the reduced manifold as a map accepting not only $\boldsymbol{q}$ but also $\boldsymbol{x}$ as an input. CROM outperforms discretization-dependent ROM approaches, such as POD and autoencoders, in terms of accuracy and memory consumption for equal latent space dimension, and in the ability to dynamically adapt the discretization during time integration.

While offering key advantages over prior ROM methods, CROM also inherits a limitation of ROM. It can only treat PDE solutions in the space spanned by the manifold-parameterization function (determined by the training data) and does not generalize to arbitrary unseen scenarios. Such a limitation is also commonly found among other implicit neural representation works (Xie et al., 2021). Future research may consider improving generalizability and data efficiency via meta-learning and integration of stronger priors (Sitzmann et al., 2020a).

Compared to end-to-end ML solutions to PDEs (Sanchez-Gonzalez et al., 2020), CROM employs the neural network strictly as a spatial representation (see Sections 4.1 and 4.3) for the kinematics and solves the PDE using classical PDE-integration numerical methods (see Section 4.2). As such, we believe CROM will open doors for more forthcoming hybrid ML-PDE solutions. As shown in our work, these solutions can retain the PDE's physical invariants (see Figure 4), allow for easy integration with existing PDE solvers (see Section 4.2), and obtain practical computational savings that can be directly employed in production (see Figure 9).

ACKNOWLEDGMENTS

This work was supported in part by the National Science Foundation (Grant CBET-17-06689), Meta, Natural Sciences and Engineering Research Council of Canada (Discovery Grant), and SideFX. We thank Honglin Chen and Rundi Wu for their insightful discussions. We thank Rundi Wu for sharing his implementation of Stable Fluids. We thank Raymond Yun Fei for rendering advice. We thank Keenan Crane for the "Origins of the Pig" mesh.

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

# Appendix

We highly encourage viewing the **supplementary video** where our results are best illustrated.

## A  ADDITIONAL LITERATURE REVIEW

### A.1  REDUCED-ORDER MODELING

Accelerating PDE solutions via reduced-order modeling (ROM) has a rich history. Two common building blocks of ROM are (1) identifying a low-dimensional latent space of the original complex system and (2) solving PDEs via latent space dynamics, i.e., evolving the latent space vector over time.

Early works on latent space construction started with linear methods where the the latent space and the full space has a linear relationship. Notable works include proper orthogonal decomposition (Berkooz et al., 1993; Holmes et al., 2012; Barbič & James, 2005; Barbič, 2012), the reduced-basis technique (Prud'homme et al., 2002; Rozza et al., 2007), balanced truncation (Moore, 1981), rational interpolation (Baur et al., 2011; Gugercin et al., 2008), and Craig–Bampton model reduction (Craig Jr & Bampton, 1968). Recently, nonlinear methods (Hartman & Mestha, 2017; Kashima, 2016; Gu, 2011; Erichson et al., 2019; Maulik et al., 2020; Regazzoni et al., 2019; Fulton et al., 2019; Maulik et al., 2021; Romero et al., 2021; Shen et al., 2021), often constructed via neural networks, have gained significant attentions due to their ability to more accurately construct the manifold-parameterization function. In particular, nonlinear manifolds significantly outperform their linear counterparts on slowly decaying Kolmogorov n-width problems (e.g., advection) (Peherstorfer, 2022; Ohlberger & Rave, 2013; Peherstorfer & Willcox, 2015; Taddei et al., 2015; Peherstorfer, 2020; Ehrlacher et al., 2020; Lee & Carlberg, 2021).

Most of these prior works exclusively focus on building a latent space for the *already-dicretized* vector fields of interest. Chen et al. (2023); Pan et al. (2023) and our work are the first attempts at constructing the latent space for the continuous vectors themselves, thereby allowing the training data to come from any discretizations.

After the latent space is identified, solving PDEs with parameters unseen during training requires evolving the latent space vector over time. Two families of approaches have been proposed for latent space dynamics. The first family of approaches (Kim et al., 2019) computes latent space dynamics via a data-driven operator that is learned from the training data. A key benefit from learning the operator from data is that the user requires no knowledge of the system's governing equations and does not need access to the training solver's source code. However, the downside of this approach is that additional assumptions (e.g., the availability of a Koopman invariant subspace as in Lusch et al. (2018) and the existence of a finite-dimensional quadratic representation as it is in the lift and learn approach by Qian et al. (2020)) are made. Without these assumptions, the learned operator is prone to violation of physical laws. The second family of approaches (Lee & Carlberg, 2020; Fulton et al., 2019), to which our method belongs, computes latent space dynamics according to the PDE used for training data generation. The advantage of this approach is that it does not require learning an additional time-evolution operator. It works with arbitrary dynamical systems with known PDE governing equations and conserves physical laws by construction since explicit PDE model is employed. However, the disadvantage is that the PDE model must be known.

## B  ROBUST SAMPLING

To select the integration samples robustly, we aim to balance the computation speed and accuracy. CROM employs $|\mathcal{M}|$ spatial samples, and the computation speed roughly scales linearly with the number of samples. To maximize computation speed, we aspire to select the fewest number of samples possible.

Given a target accuracy, our greedy algorithm selects the fewest number of integration samples from the discrete samples $\{\boldsymbol{x}^i\}_{i=1}^{P}$ of the full-order PDE solution in order to achieve the target accuracy.

---

**Algorithm 1:** Robust sampling

---

1   $\mathcal{M} = \{\boldsymbol{x}^k\}$ // randomly initialize integration sample set, $k \in \{1, \ldots, P\}$
2   **while** *true* **do**
3     $res\_vec = Calculate\_Residual(\mathcal{M})$
4     **if** $Metric(res\_vec) < target\_accuracy$ **then**
5       **return** $\mathcal{M}$
6     **else**
7       $indices = Max\_Indices(res\_vec, \mathcal{Q})$ // find $\mathcal{Q}$ indices with the largest individual residuals
8       $res\_list = \boldsymbol{0} \in \mathbb{R}^{\mathcal{Q}}$
9       **for** $i$ *in indices* **do**
10        $res\_list[i] = Metric(Calculate\_Residual(\mathcal{M} \cup \{\boldsymbol{x}^i\}))$
11       **end**
12       $j = Min\_Indices(res\_list, 1)$ // find the index with the smallest global residual
13       $\mathcal{M} = \mathcal{M} \cup \{\boldsymbol{x}^j\}$
14     **end**
15 **end**
16 **Function** $Calculate\_Residual(\mathcal{M})$**:**
17     $res\_vec = \boldsymbol{0} \in \mathbb{R}^{P}$
18     **for** $\boldsymbol{\mu} \in \mathcal{D}_{\text{train}}$ **do**
19       Obtain the final latent space vector $\boldsymbol{q}(t_T; \boldsymbol{\mu})$ with the integration samples $(\mathcal{M})$
20       **for** $i = 1, \ldots, P$ **do**
21        $res\_vec[i] + = \|\boldsymbol{g}_{\theta_g}(\boldsymbol{x}^i, \boldsymbol{q}(t_T; \boldsymbol{\mu})) - \boldsymbol{f}(\boldsymbol{x}^i, t_T)\|$
22       **end**
23     **end**
24     **return** $res\_vec$
25 **Function** $Metric(res\_vec)$**:**
26     **return** $mean(res\_vec) + max(res\_vec)$

---

In every iteration, the greedy algorithm adds one spatial sample to the sample set $\mathcal{M}$ and lowers the error of latent space dynamics. Specifically, the algorithm loops over $\mathcal{Q}$ spatial samples with the largest individual residuals. We test adding each sample to the integration sample set and compute a new global residual. From these $\mathcal{Q}$ samples, we select the one leading to the smallest new global residual and add it to the actual sample set. The algorithm repeats until the target accuracy is met. In practice, we find $\mathcal{Q} = 10$ gives sufficient results. Algorithm 1 describes the details. Compared to the naive uniform sampling approach, our robust sampling scheme yields significantly more accurate results (see Figure 4). We also highlight that Algorithm 1 happens before any simulation occurs (latent space dynamics). It is strictly a pre-computation.

## C   NETWORK INVERSION

In order to generate latent space dynamics, we need to invert the neural network employed for the manifold-parameterization function. We do so by solving the minimization problem:

$$\min_{\boldsymbol{q}_{n+1} \in \mathbb{R}^r} \sum_{\boldsymbol{y} \in \mathcal{M}} \|\boldsymbol{g}_{\theta_g}(\boldsymbol{y}, \boldsymbol{q}_{n+1}) - \boldsymbol{f}(\boldsymbol{y}, t_{n+1})\|_2^2. \tag{8}$$

Since this is a nonlinear least-squares problem, we solve it with the classic iterative Gauss-Newton algorithm (Nocedal & Wright, 2006). Unlike Newton's method, Gauss-Newton does not require the Hessian of the objective function. Consequently, in addition to the neural network itself, we only need to evaluate the gradient of the neural network with respect to the latent space vector $\frac{\partial \boldsymbol{g}_{\theta_g}}{\partial \boldsymbol{q}}(\boldsymbol{y}, \boldsymbol{q}_{n+1})$. We also equip our Gauss-Newton implementation with a standard backtracking line-search scheme to improve the convergence rate.

### C.1 INITIAL GUESS FOR THE NONLINEAR SOLVER

An initial guess close to the minimum is necessary for the convergence of Gauss-Newton. The previous time step latent space vector $q_n$ serves as an ideal initial guess since $f$ does not vary much over a time step. However, in the extreme cases where $f$ deviates significantly over one time step, we find it helpful to compute the initial guess through an encoder network. For convenience, we employ the encoder network from training, which requires computing the dynamics for $P$ samples. Future work should consider building this encoder network strictly for the degrees of freedom in $\mathcal{M}$. We can further facilitate nonlinear solver convergence by encouraging smoothness of the latent space, such as employing a Jacobian penalty term (Chen et al., 2023) / local isometry term (Du et al., 2021) and enforcing Lipschitz constraints (Liu et al., 2022). These smoothness regularizations would be particularly important for challenging PDE cases involving bifurcations and buckling (Brush et al., 1975).

### C.2 LINEARIZATION

We can further obtain computation-save by bypassing the iterative nonlinear solver and linearizing the least-squares problem.

We define $\Delta f$ as,

$$\Delta f(y, t_{n+1}) = f(y, t_{n+1}) - f(y, t_n) \tag{9}$$
$$= f(y, t_{n+1}) - g_{\theta_g}(y, q_n). \tag{10}$$

Employing Taylor expansion of $g_{\theta_g}$ at $q_n$, we have $g_{\theta_g}(y, q_{n+1}) \approx g_{\theta_g}(y, q_n) + \frac{\partial g_{\theta_g}}{\partial q}(y, q_n)\Delta q_{n+1}$, where $q_{n+1} = q_n + \Delta q_{n+1}$. Therefore, the original objective function can be approximated as,

$$g_{\theta_g}(y, q_{n+1}) - f(y, t_{n+1}) = g_{\theta_g}(y, q_{n+1}) - g_{\theta_g}(y, q_n) - \Delta f(y, t_{n+1}) \tag{11}$$
$$\approx \frac{\partial g_{\theta_g}}{\partial q}(y, q_n)\Delta q_{n+1} - \Delta f(y, t_{n+1}). \tag{12}$$

Therefore, we can obtain the latent space dynamics by solving the *linear* least-squares problem,

$$\min_{\Delta q_{n+1} \in \mathbb{R}^r} \sum_{y \in \mathcal{M}} \|\frac{\partial g_{\theta_g}}{\partial q}(y, q_n))\Delta q_{n+1} - \Delta f(y, t_{n+1})\|_2^2. \tag{13}$$

This is the normal equation and can be solved in the closed form:

$$\Delta q_{n+1} = (J^T J)^{-1} J^T b, \tag{14}$$

where $J$ is the $(|\mathcal{M}| \cdot d)$ by $r$ Jacobian matrix that contains $\frac{\partial g_{\theta_g}}{\partial q}(y, q_n), \forall y \in \mathcal{M}$; $b$ is the $(|\mathcal{M}| \cdot d)$ by 1 residual vector containing $\Delta f(y, t_{n+1}), \forall y \in \mathcal{M}$.

Since our work features both small $|\mathcal{M}|$ and small $r$, the closed-form network inversion costs just a few small dense matrix multiplications and thereby introduces minimal overhead. In fact, the solver itself is extremely efficient, and the majority of the computation cost is forming the Jacobian matrix $J$ itself, whose optimization will be discussed in Appendix E.1.1. Additional performance details are listed in Appendix N.

Although our network inversion step follows the standard optimal-projection-based ROM literature (Carlberg et al., 2013), future work may consider directly learning an inverse during training time. The encoder introduced in the manifold construction process may be a step towards learning an inverse. However, the encoder in its current form does not have optimal accuracy and is bounded to a particular discretization. Future work should consider lifting these limitations.

## D    Network Details

Consistent with the implicit neural representation literature, we parameterize the manifold-parameterization function manifold with an MLP network $\boldsymbol{g}_{\theta_g}$. The input dimension of the network is $m + r$ while the output dimension of the network is $d$, where $m$ is the dimension of the input spatial vector while $d$ is the output dimension of the vector field of interest. Our MLP network contains 5 hidden layers, each of which has a width of $(\beta \cdot d)$, where $\beta$ is the hyperparameter that defines the learning capacity of the network. Essentially, our network has two tunable hyperparameters, $r$ and $\beta$. A detailed study on these hyperparameters will be discussed in Appendix F.

Since we require the network to be continuously differentiable with respect to both the spatial coordinates $\boldsymbol{x}$ and the latent space vector $\boldsymbol{q}$, we adopt continuously differentiable activation functions. In practice, either ELU (Clevert et al., 2015) or SIREN (Sitzmann et al., 2020b) serves this purpose.

For the encoder network $\boldsymbol{e}_{\theta_e}$, the input is a 1D vector of length $P$ with $d$ channels. The output is a vector of dimension $r$. Using MLPs would lead to enormous network size when $P$ is large. We therefore design an encoder network first to apply multiple 1D convolution layers of kernel size 6, stride size 4, and output channel size $d$ until the output 1D vector's length is the closest to $32/d$ but no smaller. Afterward, the encoder network reshapes the vector into 1 channel and applies an MLP layer to reduce the dimension of the vector to 32. The last MLP layer then transforms the previous 32-dimensional vector into dimension $r$. Discretization-specific encoder networks can also be used, e.g., PointNet for point clouds and convolutional neural network for grid data.

### D.1    Training Details

We use the Adam optimizer (Kingma & Ba, 2014) for stochastic gradient descent. We use the Xavier initialization for ELU layers and the default initialization ($\omega_0 = 30$) for SIREN layers. Unless otherwise noted, we train with a base learning rate of $lr = 1e - 4$ and adopt a learning rate decay strategy ($10 \cdot lr \rightarrow 5 \cdot lr \rightarrow 2 \cdot lr \rightarrow 1 \cdot lr \rightarrow 0.5 \cdot lr \rightarrow 0.2 \cdot lr$). For each aforementioned learning rate, we train for $30,000$ epochs. We adopt a batch size 16 (i.e., 16 simulation snapshots) for the encoder and therefore a batch size of $16 \cdot P$ for the implicit-neural-representation-based manifold-parameterization function. We implement the entire training pipeline in PyTorch Lightning(Falcon et al., 2019) which facilitates distributed training across multiple GPUs. The training vector fields are standardized to have zero mean and unit variance. The training spatial coordinates are also standardized for the ELU implementation, but they are preprocessed to be between $[-1, 1]$ for the SIREN implementation.

## E    Network Gradients

Gradients of the implicit neural representation ($\boldsymbol{\nabla}_{\boldsymbol{x}}\boldsymbol{g}_{\theta_g}$ and $\frac{\partial \boldsymbol{g}_{\theta_g}}{\partial \boldsymbol{q}}$) are crucial for our reduced-order pipeline. We use them to compute the spatial and temporal gradients of the vector fields, $\boldsymbol{\nabla}\boldsymbol{f}(\boldsymbol{x}, t) = \boldsymbol{\nabla}_{\boldsymbol{x}}\boldsymbol{g}_{\theta_g}(\boldsymbol{x}, \boldsymbol{q})$ and $\dot{\boldsymbol{f}}(\boldsymbol{x}, t) = \frac{\partial \boldsymbol{g}_{\theta_g}}{\partial \boldsymbol{q}}(\boldsymbol{x}, \boldsymbol{q})\dot{\boldsymbol{q}}$. Furthermore, the gradient with respect to the latent space $\frac{\partial \boldsymbol{g}_{\theta_g}}{\partial \boldsymbol{q}}$ is also a key ingredient for the network inversion (Appendix C).

### E.1    Network Gradients via Direct Differentiation

One way to compute the gradients is through directly differentiating the continuously differentiable network.

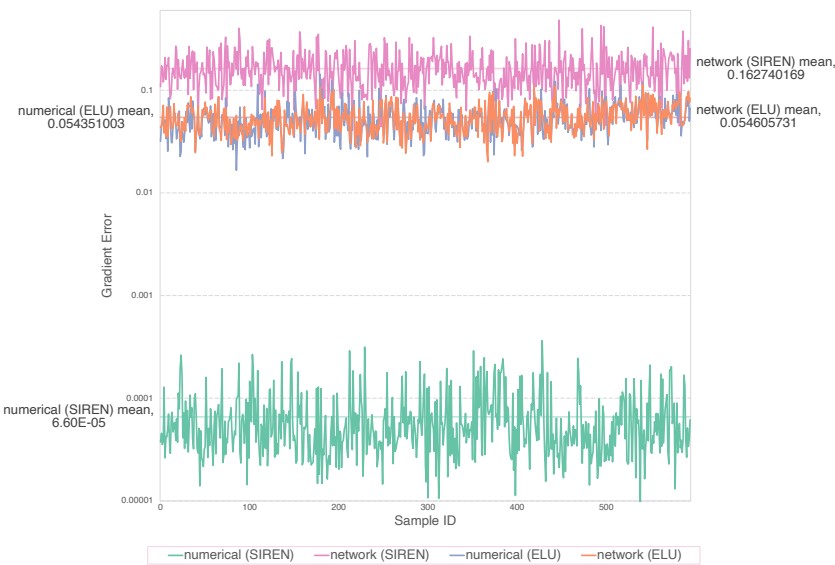

Figure 11: Gradient errors evaluated at the centers of tetrahedra. Numerical differentiation (specifically, finite element with linear basis) significantly improves the gradient accuracy of SIREN.

### E.1.1 EFFICIENT IMPLEMENTATION

---

**Algorithm 2:** Network gradient

---

**Input:** $\boldsymbol{x}$

**Output:** $\frac{\partial \boldsymbol{g}_{\theta_g}}{\partial \boldsymbol{x}}(\boldsymbol{x}), \boldsymbol{g}_{\theta_g}(\boldsymbol{x})$

1 Let $\boldsymbol{g}_{\theta_g} = \boldsymbol{g}_n \circ \cdots \circ \boldsymbol{g}_1$ be a $n$-layer MLP network of interest.

2 Initialization: $\frac{\partial \boldsymbol{y}}{\partial \boldsymbol{x}} = \boldsymbol{I}, \boldsymbol{y} = \boldsymbol{x}$

3 **for** $i = 1, \ldots, n$ **do**

4 $\quad \frac{\partial \boldsymbol{y}}{\partial \boldsymbol{x}} = \frac{\partial \boldsymbol{g}_i}{\partial \boldsymbol{x}}(\boldsymbol{y})\frac{\partial \boldsymbol{y}}{\partial \boldsymbol{x}}$

5 $\quad \boldsymbol{y} = \boldsymbol{g}_i(\boldsymbol{y})$

6 **end**

7 **return** $\frac{\partial \boldsymbol{y}}{\partial \boldsymbol{x}}, \boldsymbol{y}$

---

Unfortunately, computing these differentiations via auto-diff (computational graph tracking) is too slow for the high-performance application explored in this work. To ensure maximum efficiency, we implement the gradients analytically through the chain rules of each network layer, similar to the grad net approach by (Lindell et al., 2021). Since we only use fully-connected layers, the implementation is straightforward. In addition, along the way of computing the gradient via chain rules, we also obtain the function value of the neural network itself. Therefore, we obtain both the gradient and the function value in a single forward pass. Algorithm 2 describes the details.

### E.1.2 FAILURE MODES

While direct differentiation is generally accurate, we observe failure modes when the training samples are sparse. Consider the case where $\boldsymbol{g}_{\theta_g}(\boldsymbol{x}, \boldsymbol{q}) = \boldsymbol{x}$, e.g., the undeformed, reference configuration of elastodynamics. In this case, the ground truth spatial gradient is $\nabla_{\boldsymbol{x}} \boldsymbol{g}_{\theta_g} = \boldsymbol{I}$.

We train the neural network with spatial samples discussed in Appendix L.4.1. After training, we evaluate the spatial gradients at the center of each tetrahedron and compute its error (from the identity matrix) in the L2 norm. As shown in Figure 11, errors are observed in both ELU and SIREN networks while SIREN's error is significantly larger, c.f.,(Yang et al., 2021). This discrepancy can be understood as the high-frequency prior by SIREN being more suitable for approximating

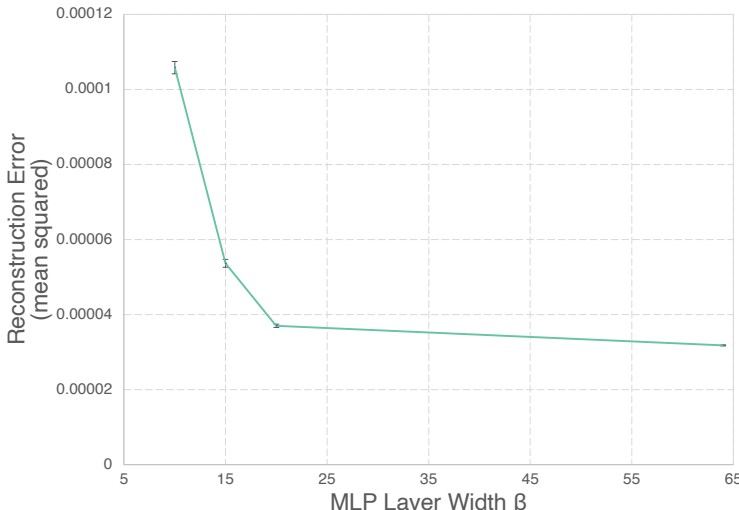

Figure 12: Reconstruction accuracy vs. the MLP layer width ($\beta$). For each setup, we repeat the training for $8$ times. The mean values and the error bars are shown. Increasing MLP layer width leads to higher accuracy.

high-frequency functions. By contrast, the low-frequency prior by ELU is better at approximating low-frequency functions (Hertz et al., 2021).

We find training with more samples and/or gradient supervision (Chen et al., 2023) reduces the errors above at the cost of additional training resources but does not resolve the issue completely. Future work may consider more advanced gradient regularization methods (Liu et al., 2022).

### E.2 NETWORK GRADIENTS VIA NUMERICAL DIFFERENTIATION

Another approach to computing neural network gradients is numerical approximation, such as the finite difference method and the finite element method. Figure 11 demonstrates the drastically improved gradient accuracy with the finite element method (linear basis function) on networks trained with SIREN. We find such a hybrid approach to be an ideal middle ground, i.e., we represent the function itself with the discretization-independent neural network and compute the gradient via discretization-dependent numerical differentiation.

With numerical gradients, users still benefit from our continuous representation that allows them to use *any* discretization for gradient computation (e.g., ground impact elasticity deformation with remeshing). Prior ROM approaches require these numerical gradients to be computed by a fixed mesh.

### F HYPERPARAMETERS

During the offline training stage, we can modify two network architecture hyperparameters: the size of the latent space ($r$) and the width of the MLP ($\beta$). After training and during the online deployment stage of the model, the key hyperparameter is the number of integration samples ($|\mathcal{M}|$). In this section, we run a sensitivity analysis on these hyperparameters.

Figure 12 shows that increasing the width of each MLP layer improves the reconstruction accuracy. Unless otherwise noticed, all reconstruction errors reported are mean squared errors (MSE). While Figure 12 demonstrates an empirical convergence, future work should consider a theoretical analysis (Kunisch & Volkwein, 2002).

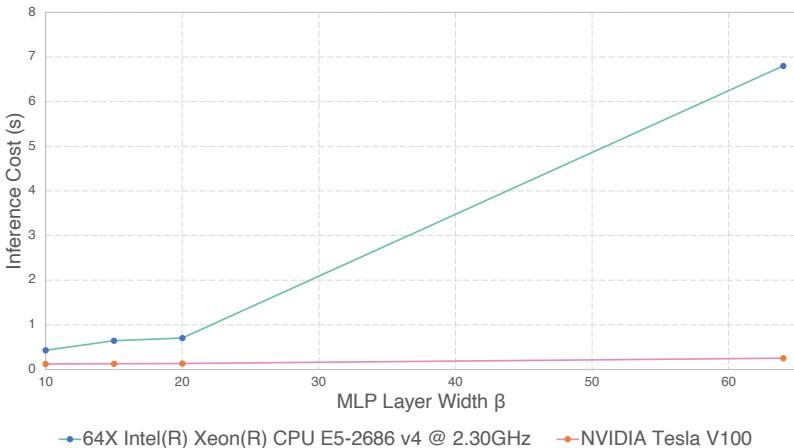

Figure 13: Network inference cost (CPU and GPU) vs. the MLP layer width ($\beta$). Inference cost (over 100 runs) increases as the network size increases. CPU suffers from a more severe performance drop than GPU.

However, large layer widths also entail higher computational cost (see Figure 13). Compared to the highly parallelized GPUs, CPUs suffer from a more severe performance drop due to the larger network size.

Unlike traditional PCA-based model reduction approaches (e.g., POD), increasing the dimension of the latent space vector does not yield higher accuracy for our approach (see Figure 10). In prior ROM approaches, augmenting the dimension of the latent space vector directly increases the number of entries in the manifold-parameterization function matrix, which, in turn, improves the approximation capacity of the manifold-parameterization function. However, as shown in Figure 12, in the implicit neural representation framework, we can control the learning capacity of the network architecture by explicitly modifying the width of the MLP layer. Therefore, the importance of the latent space dimension diminishes. This experimental observation is consistent with prior work's theoretical analysis (Remark 2.1 by Lee & Carlberg (2020)) which says the lower bound of the latent space dimension is the dimension of the problem parameters (e.g., material properties) plus one. Consequently, one should consider increasing the learning capacity of the manifold-parameterization function instead of increasing the latent space dimension.

During the online stage of the reduced-order model, we select integration samples via the greedy sampling scheme (Appendix B). Increasing the number of integration samples leads to higher accuracy (Figure 14).

## G  THERMODYNAMICS

### G.1  CONTINUOUS PDE

$$\frac{\partial u}{\partial t} - \nu(x)\frac{\partial^2 u}{\partial x^2} = 0 \tag{15}$$

In thermodynamics, we study the 1D heat equation of the spatiotemporal dependent temperature $u$. We assume a zero-Dirichlet boundary condition, though other boundary conditions can also be incorporated.

### G.2  FULL-ORDER MODEL

The full-order model discretizes the spatial vector field with a regular Eulerian grid using 501 equally spaced samples ($P = 501$).

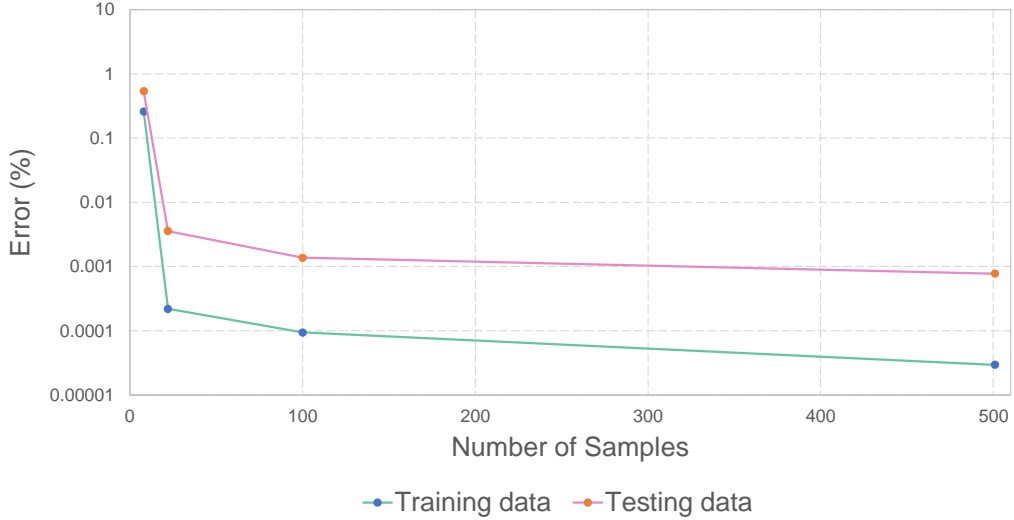

Figure 14: PDE solution accuracy vs. the number of samples. Increasing integration sample counts yields higher accuracies on both training and testing data.

We then approximate the spatial gradient using the finite difference method,

$$\frac{\partial^2 u}{\partial x^2}(x^i, t_n) = \frac{u(x^i - \Delta x, t_n) + u(x^i + \Delta x, t_n) - 2u(x^i, t_n)}{\Delta x^2}, \tag{16}$$

where $\Delta x$ is the grid spacing.

We use this to compute the next-time step velocity,

$$\dot{u}_{n+1}^i = \nu(x^i)\frac{\partial^2 u}{\partial x^2}(x^i, t_n). \tag{17}$$

We assume a first-order explicit time-stepping scheme,

$$u_{n+1}^i = u_n^i + \Delta t\dot{u}_{n+1}^i. \tag{18}$$

### G.3 REDUCED-ORDER MODEL

Instead of a finite-difference treatment of the spatial gradient, we compute the gradient via direct differentiation of the network. The latent space dimension is $r = 16$ and the width of the MLP is $\beta = 128$.

### G.4 TRAINING AND TESTING DATA

We generate training data by setting $\nu$ to different piecewise constant functions of three regions, i.e., the parameter vector $\boldsymbol{\mu} \in \mathcal{D} = [0.2, 1.0]^3 \subset \mathbb{R}^3$. Figure 15 displays the initial and the final states of training data. Notice how different regions diffuse with different speeds due to the spatially varying $\nu$. In total, 8 PDE temporal sequences (of 100 time steps) with different $\nu$'s are generated for training. We sample another 4 $\nu$'s for testing purposes. All training and testing data adopt the same initial condition (Figure 15a).

### G.5 RESULTS

Figure 16 demonstrates the performance of the reduced-order model on the testing dataset. With the robust sampling scheme, we can achieve $0.43\%$ error using just 22 spatial samples (see Figure 4). In addition, by incorporating different initial conditions (e.g., Gaussian profiles centered at different horizontal locations) in the training data, our approach can capture diffusion with a wide range of initial conditions (Figure 17).

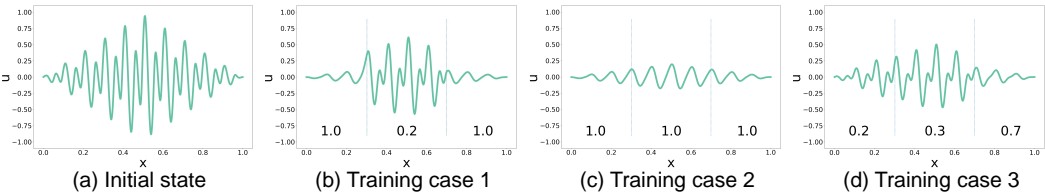

Figure 15: Thermodynamics: training data.

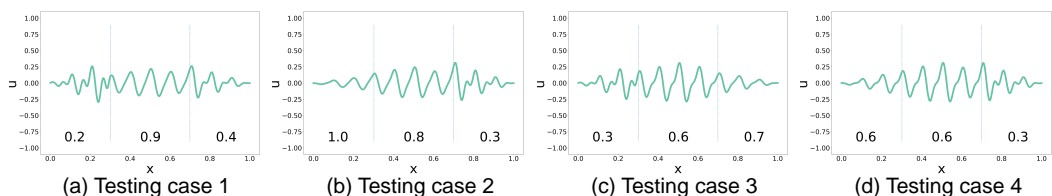

Figure 16: Thermodynamics: reduced-order simulation on the testing dataset.

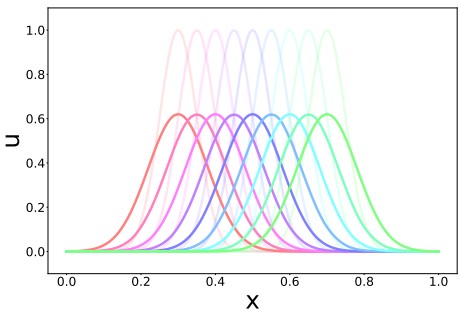

Figure 17: Our reduced-order diffusion solver with various initial conditions (transparent) and the final diffused states. Different colors correspond to different initial conditions.

To gauge our approach's ability to conserve physical laws, we measure the energy evolution over time in the region between $x_1 = 0.396$ and $x_2 = 0.526$ (Figure 16). As shown by Cannon (1984), the escaped energy (EE) is measured by the flux difference ($\frac{\partial u}{\partial x}(x_1, t) - \frac{\partial u}{\partial x}(x_2, t)$) at the left and right boundaries integrated over time; the stored energy (SE) is measured by the heat energy density, proportional to the temperature ($u$), integrated over space; the total energy (TE) is the sum of the prior two energies, i.e., TE=EE+SE. Figure 4 demonstrates that the reduced simulation preserves the total energy over time.

## H    IMAGE PROCESSING

### H.1    CONTINUOUS PDE

$$\frac{\partial u}{\partial t} - \nu(\boldsymbol{x})\boldsymbol{\nabla}^2 u = 0 \tag{19}$$

In image processing, we study the Perona–Malik diffusion equation of the gray-scale image $u$ (Perona & Malik, 1990).

### H.2    FULL-ORDER MODEL

The full-order model discretizes the spatial 2D pixel field with a regular Eulerian grid using $256 \times 256$ equally spaced samples ($P = 65,536$).

We then approximate the spatial gradient using the finite difference method,

$$\frac{\partial^2 u}{\partial x^2}(x^i, y^i, t_n) = \frac{u(x^i - \Delta x, y^i, t_n) + u(x^i + \Delta x, y^i, t_n) - 2u(x^i, y^i, t_n)}{\Delta x^2}, \tag{20}$$

$$\frac{\partial^2 u}{\partial y^2}(x^i, y^i, t_n) = \frac{u(x^i, y^i - \Delta y, t_n) + u(x^i, y^i + \Delta y, t_n) - 2u(x^i, y^i, t_n)}{\Delta y^2}. \tag{21}$$

We use these to compute the next-time step velocity,

$$\dot{u}^i_{n+1} = \nu(x^i, y^i)(\frac{\partial^2 u}{\partial x^2}(x^i, y^i, t_n) + \frac{\partial^2 u}{\partial y^2}(x^i, y^i, t_n)). \tag{22}$$

We assume a first-order explicit time-stepping scheme,

$$u^i_{n+1} = u^i_n + \Delta t \dot{u}^i_{n+1}, \tag{23}$$

where $\Delta x = \Delta y$ is the grid spacing.

### H.3    REDUCED-ORDER MODEL

The reduced-order model follows the same treatments as Appendix G.3.

### H.4    TRAINING AND TESTING DATA

Similar to Appendix G.4, we generate training data by setting $\nu$ to different piecewise constant functions of four equally-spaced regions, i.e., the parameter vector $\boldsymbol{\mu} \in \mathcal{D} = \{0, 0.2\}^4 \subset \mathbb{R}^4$. $\mathcal{D}_{\text{train}}$ contains 11 parameter vectors (Figure 18), where 0, 2 or 3 regions have zero diffusion coefficients, i.e., no blurring. $\mathcal{D}_{\text{test}}$ contains 4 parameter vectors, where only 1 region is unblurred. For each training and testing parameter vector, we run a simulation of 20 time steps. All training and testing data adopt the same initial condition (Figure 18a).

### H.5    RESULTS

Figure 19 demonstrates the reduced simulation's performance on the testing dataset. Furthermore, with the robust sampling scheme, we can achieve 36.1 PSNR using just 63 spatial samples (see Figure 20). The dimension reduction is 99.90%.

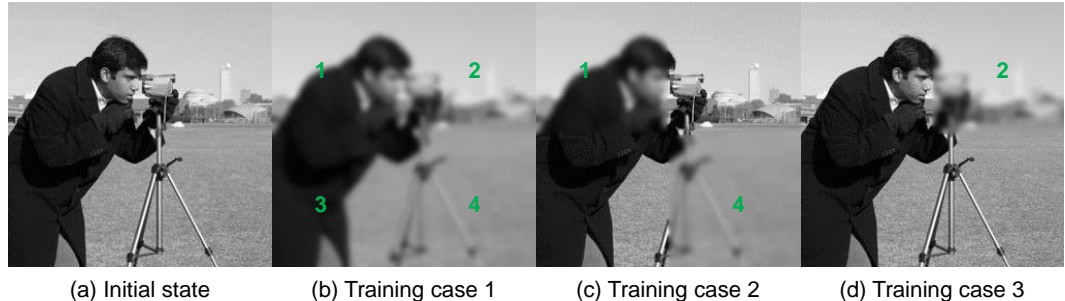

(a) Initial state     (b) Training case 1     (c) Training case 2     (d) Training case 3

Figure 18: Image Processing:training dataset. The blurred region is numbered.

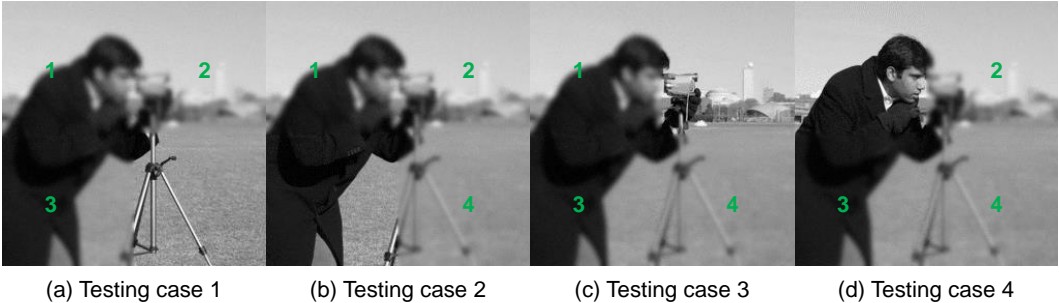

(a) Testing case 1     (b) Testing case 2     (c) Testing case 3     (d) Testing case 4

Figure 19: Image Processing: our reduced-order simulation successfully blurs the target regions in the testing dataset. The blurred region is numbered.

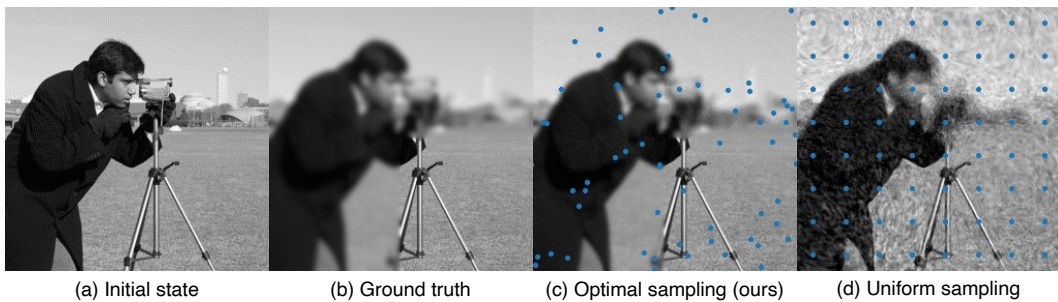

(a) Initial state     (b) Ground truth     (c) Optimal sampling (ours)     (d) Uniform sampling

Figure 20: Blurring of portions of the image (20 time steps). (b) Ground truth solution uses all $P = 65,536$ pixels. (c) Our approach uses very few integration samples ($|\mathcal{M}| = 63$, blue circles) and obtains a similar result as the ground truth (PSNR 36.1). (d) Naive uniform sampling ($|\mathcal{M}| = 64$, blue circles) leads to a poor agreement with the ground truth (PSNR 23.4).

# I ADVECTION EQUATION

Nonlinear ROM techniques have been shown to significantly outperform linear ROM techniques on problem with slowly decaying Kolmogorov n-widths (Peherstorfer, 2022; Lee & Carlberg, 2020). In this section, we test CROM, also a nonlinear ROM method, on a transport-dominated problem, given by the advection equation.

## I.1 CONTINUOUS PDE

$$\frac{\partial u}{\partial t} + (a \cdot \nabla)u = 0, \tag{24}$$

where $u$ is the advected quantity and $a$ is the advection velocity.

## I.2 FULL-ORDER MODEL

The full-order model discretizes the 1D spatial field with a regular grid using $100$ equally spaced samples. The PDE is temporally discretized via forward Euler.

## I.3 REDUCED-ORDER MODEL

We vary the latent space dimension $r$ and use a fixed width of the MLP $\beta = 20$.

## I.4 TRAINING AND TESTING DATA

In this example, we consider a reproductive case where the training and the testing data are the same. In particular, the initial condition is given by a Gaussian profile and is then advected under a constant velocity.

## I.5 RESULTS

Since this example is reproductive and the only parameter of the system is time $t$, the intrinsic solution-manifold dimension (Lee & Carlberg, 2020) is $1$. Figure 21 demonstrates that CROM indeed accurately captures the transport behavior of the Gaussian profile even at the the intrinsic solution-manifold dimension ($r = 1$). By contrast, POD suffers from serious artifacts with lower dimensional latent space and requires a significantly higher dimensional latent space ($r = 16$) in order to model the advection behavior. Furthermore, Figure 6 shows that our approach is also more accurate than the convolutional autoencoder approach by Lee & Carlberg (2020) when $r = 1$.

# J BURGERS' EQUATION

In this section, we continue to investigate problems with slowly decaying Kolmogorov n-widths. In particular, we obtain the setup by Lee & Carlberg (2020) (see Section 7.1 of their paper) and compare with their (discretization-dependent) convolutional autoencoder manifold.

## J.1 CONTINUOUS PDE

$$\frac{\partial w}{\partial t} + \frac{\partial 0.5 w^2}{\partial x} = 0.02 e^{\mu_D x}, \tag{25}$$

where $w$ is the variable of interest. The initial condition is as follows.

$$w(0,0) = 4.25 \tag{26}$$
$$w(x,0) \quad \forall x \in (0, 100] \tag{27}$$

We further assume a zero Neumann boundary condition.

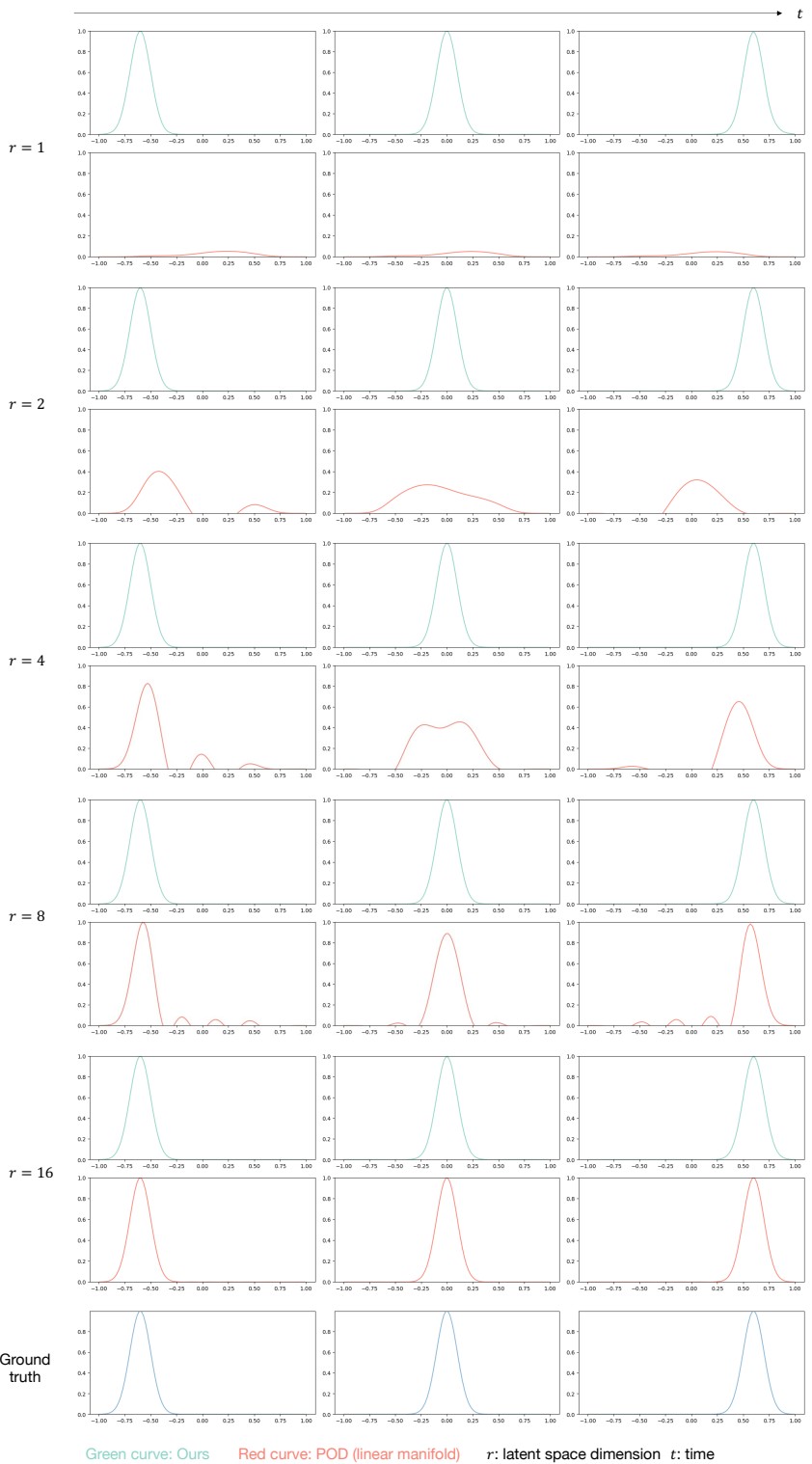

Figure 21: Advection. Our CROM employs a nonlinear manifold-parameterization function and accurately captures the advection phenomenon with just a one-dimensional latent space. By contrast, linear manifolds, such as POD, require a significantly larger latent space dimension. Lower dimensional latent spaces on linear manifolds yield significant artifacts due to the inability to model slowly decaying Kolmogorov n-widths problems.

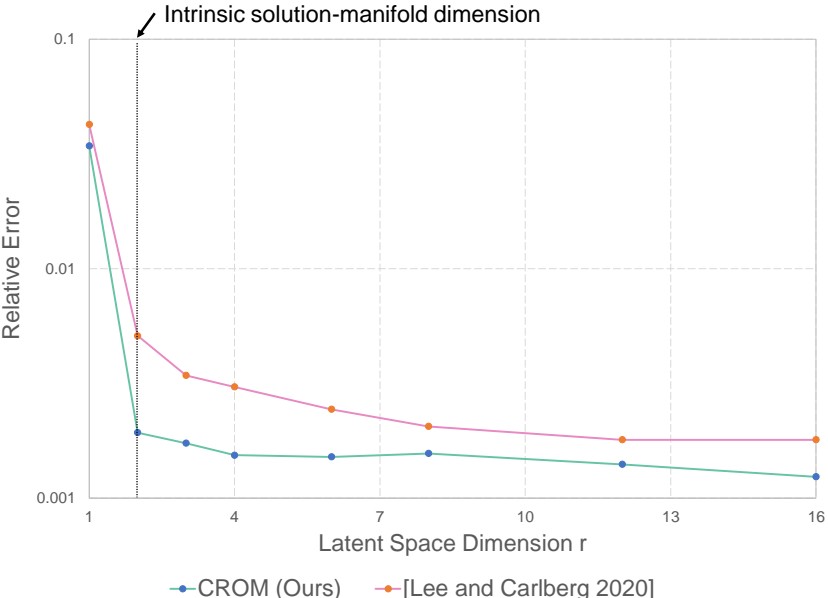

Figure 22: Reconstruction errors. Our approach yields higher accuracies than the convolutional autoencoder approach by Lee & Carlberg (2020) across different latent space dimensions. Our method's accuracy advantage is particularly noticeable at the intrinsic solution-manifold dimension (see the vertical dashed line).

## J.2 FULL-ORDER MODEL

The full-order model discretizes the 1D spatial field with a regular grid using 256 equally spaced samples. The PDE is temporally discretized via forward Euler.

## J.3 REDUCED-ORDER MODEL

We vary the latent space dimension $r$ and use a fixed width of the MLP $\beta = 64$.

## J.4 TRAINING AND TESTING DATA

We generate training and testing data by varying the parameter $\mu_D$. We set the training parameter to $\mu_{D_{train}} = \{0.015 + (0.015/7)j\}_{j=0,1,...7}$, resulting in 8 training cases. For online testing, we set the parameter to $\mu_{D_{test}} = 0.021$.

## J.5 RESULTS

Since we can uniquely parameterize the family of training and testing data with two variables $t$ and $\mu_D$, the intrinsic solution-manifold dimension (Lee & Carlberg, 2020) is $p^* = 2$, which is the lower bound for the latent space dimension $r$.

Figure 6b reports the result for $r = p^* = 2$. While both POD and convolutional autoencoder (CAE) approach from Lee & Carlberg (2020) display clear artifacts, CROM agrees well with the ground truth. Such an agreement at the latent space dimension lower bound (i.e., $r = p^*$) shows that CROM is very close to achieving the optimal performance of *any* nonlinear trial manifold.

Furthermore, Figure 22 demonstrates the accuracy advantage of our approach over the CAE approach across different latent space dimensions.

## K    FLUID DYNAMICS

### K.1    CONTINUOUS PDE

$$\frac{\partial \boldsymbol{u}}{\partial t} + (\boldsymbol{u} \cdot \boldsymbol{\nabla})\boldsymbol{u} = -\boldsymbol{\nabla}p + \nu\boldsymbol{\nabla}^2\boldsymbol{u} + \boldsymbol{f}_{ext}, \boldsymbol{\nabla} \cdot \boldsymbol{u} = 0 \tag{28}$$

We study the incompressible Navier-Stokes equations, where $\boldsymbol{u}$ is the velocity field, $p$ is the pressure field, and $\nu$ is the viscosity. We assume there is no external force, i.e., $\boldsymbol{f}_{ext} = \boldsymbol{0}$. We assume a no-penetration boundary condition for the velocity and a zero-Dirichlet boundary condition for the pressure.

### K.2    FULL-ORDER MODEL

The full-order model assumes an operator-splitting scheme following the classic Chorin's projection method (Chorin, 1968). In particular, we follow the implementation by Stam (1999). We summarize the major ingredients and refer to his paper for details.

We sequentially apply 3 linear operators to the velocity field: diffusion, advection, and projection. From the previous time step velocity field $\boldsymbol{u}_n$, we apply the diffusion operator according to the viscosity and obtain the diffused velocity $\boldsymbol{u}_n^{\mathrm{visc}}$. Next, we employ semi-Lagrangian to obtain the advected velocity field $\boldsymbol{u}_n^{\mathrm{adv}}$. For the last projection step, we first compute the pressure $p_{n+1}$ by solving the poisson equation, $\nabla^2 p_{n+1} = \boldsymbol{\nabla} \cdot \boldsymbol{u}_n^{\mathrm{adv}}$. Afterwards, we apply the pressure gradient to obtain the divergence-free velocity $\boldsymbol{u}_{n+1}$. All operations are done on a 2D 16 by 16 Eulerian grid.

### K.3    REDUCED-ORDER MODEL

We compute latent space dynamics ($\boldsymbol{q}_{n+1}$) of the velocity field by projecting $\boldsymbol{u}_{n+1}$ onto the manifold-parameterization function $\boldsymbol{g}_{\theta_g}$. For network hyperparameters, we adopt a latent space dimension $r = 6$ and a MLP width $\beta = 10$.

### K.4    TRAINING AND TESTING DATA

We generate training data by varying the spatially-constant viscosity: $\boldsymbol{\mu} = \nu \in \mathcal{D} = [0, 0.02] \subset \mathbb{R}$. We sample 3 different viscosities from $\mathcal{D}$. For each viscosity $\nu$, we run a simulation of 50 time steps. In total, 150 time steps are used for training. For testing, we first sample 3 viscosity values from $\mathcal{D} = [0, 0.02]$. Furthermore, we also sample 3 values from the extrapolated region $\mathcal{D}^{\mathrm{extrapolate}} = [0.02, 0.1]$. Notice how the extrapolated region has a larger range (0.08) than the interpolated region (0.02). All training and testing data adopt the same initial condition (see Figure 23).

### K.5    RESULTS

Figure 23 demonstrates our method's ability to qualitatively capture fluid dynamics with different viscosities on both interpolated data as well as extrapolated data. Surrogate models that predict velocity fields directly from viscosities (e.g., using a neural network) typically cannot handle extrapolation. By contrast, our approach handles extrapolated scenarios because we only approximate the spatial representation (kinematics) with a neural network, not the PDE itself. Therefore, as long as the extrapolated cases involve vector fields that can be represented by the low-dimensional implicit neural representation, our reduced-order solver is able to evolve the latent space vector to capture them according to the PDE. However, if the extrapolated quantity lies outside of the space spanned by the manifold-parameterization function, our approach would not be able to capture the dynamics. CROM, like most ROM approaches, does not handle arbitrary extrapolation scenarios. See more discussion on generalizability at the end of the main text.

To further test the proposed method, we set up a Karman vortex street experiment. The Karman vortex street is particularly challenging as the flow bifurcates from a steady state to the periodic vortex shedding regime. Mathematically speaking, the fixed point solution undergoes a Hopf bifurcation and becomes a limit cycle (Huerre & Monkewitz, 1990). To capture such an effect, the latent space dynamics needs to be robust under extreme nonlinearity. Many neural-network-based methods, such

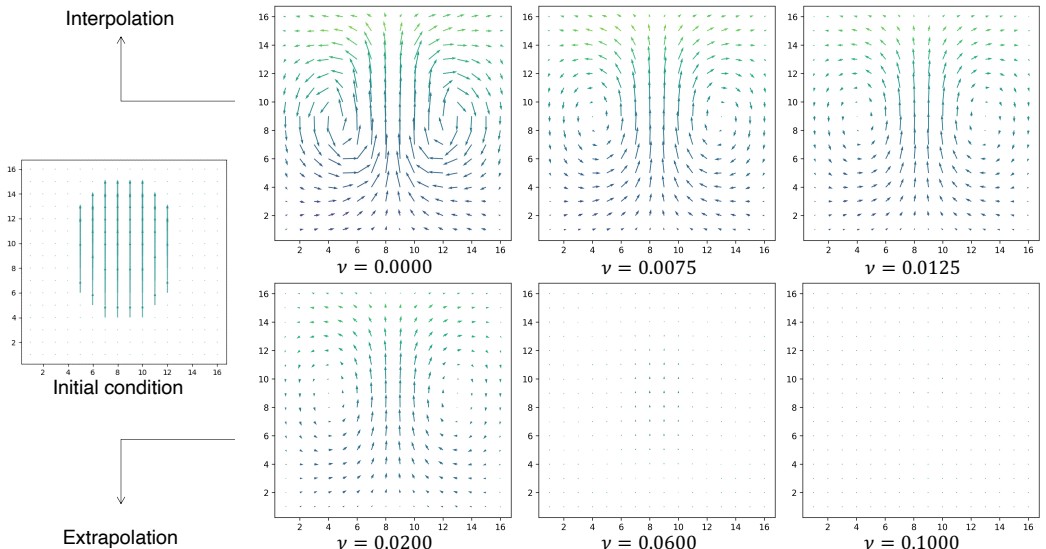

Figure 23: Fluid dynamics: our method captures the velocity fields of a wide range of viscous fluids. Large viscosity severely dissipates the velocity. All frames shown are captured from time step 25. The first row consists of the interpolated result while the second row consists of the extrapolated result well outside the training range.

as physics informed neural network (Raissi et al., 2019), are well-known to fail in this case (Chuang & Barba, 2022).

We follow the geometry and material parameters outlined by Nabizadeh et al. (2022). No hyper-reduction is considered. Figure 7 shows that our latent space dynamics method faithfully represents the ground truth flow: from the initial steady state to the bifurcated vortex shedding regime. The ground truth solver utilizes a 100 by 200 grid, with a total 40, 000 degrees of freedom. By contrast, our method captures the same level of details while employing a latent space dimension of only $r = 4$. Due to CROM's kinematics-approximation-only nature, our approach also generalize beyond the training temporal range.

Future work should test CROM on more challenging boundary conditions (Copeland et al., 2022) and chaotic systems involving non-Markovian effects (Pan & Duraisamy, 2018).

## L  SOLID MECHANICS

### L.1  CONTINUOUS PDE

We solve the elastodynamics equation arisen from solid mechanics for deformable soft body modeling:

$$\rho_0 \ddot{\phi} = \boldsymbol{\nabla} \cdot \boldsymbol{P}(\boldsymbol{\nabla}\phi) + \rho_0 \boldsymbol{B}, \tag{29}$$

where $\phi(\boldsymbol{X}, t)$ maps the undeformed (reference) position ($\boldsymbol{X}$) of an arbitrary material point from the reference configuration $\Omega$ to its deformed (current) position ($\boldsymbol{x}$) at time $t$. Proper Dirichlet and Neumann boundary conditions are applied.

We assume the first Piola–Kirchhoff stress $\boldsymbol{P}$ is strictly a function of the deformation gradient $\boldsymbol{F} = \boldsymbol{\nabla}\phi$. In particular, we adopt the corotated linear elasticity constitutive law,

$$\boldsymbol{P}(\boldsymbol{F}) = 2\mu(\boldsymbol{F} - \boldsymbol{R}) + \lambda \operatorname{tr}^2(\boldsymbol{R}^T \boldsymbol{F} - \boldsymbol{I})\boldsymbol{R}, \tag{30}$$

where $\boldsymbol{R}$ is the rotation tensor from the polar decomposition of the deformation gradient $\boldsymbol{F} = \boldsymbol{RS}$, $\lambda$ and $\mu$ are the first lame parameter (Pa) and the second lame parameter / shear modulus (Pa). The proposed framework also works with other elasticity constitutive laws and can be easily extended to capture hysteresis and inelasticity.

## L.2   Full-order Model

We adopt the tetrahedral-mesh-based linear finite element method (FEM) for the full-order model. We closely follow the course by Sifakis & Barbic (2012). Here we review only the salient features and refer to their work for theoretical and practical details.

We spatially discretize the domain of interest using linear tetrahedra. For each tetrahedron, we have four undeformed vertex positions $(\boldsymbol{X}_1, \cdots, \boldsymbol{X}_4)$ and four deformed vertex positions $(\boldsymbol{x}_1, \cdots, \boldsymbol{x}_4)$. For each vertex, we have $\boldsymbol{x}_i = \boldsymbol{F}\boldsymbol{X}_i + \boldsymbol{b}$, where $\boldsymbol{b}$ is the translation.

Consequently, we can obtain the deformation gradient $\boldsymbol{F}$ of each tetrahedron by solving $\boldsymbol{D}_s = \boldsymbol{F}\boldsymbol{D}_m$, where $\boldsymbol{D}_s = [\boldsymbol{x}_1 - \boldsymbol{x}_4 \quad \boldsymbol{x}_2 - \boldsymbol{x}_4 \quad \boldsymbol{x}_3 - \boldsymbol{x}_4]$ and $\boldsymbol{D}_m = [\boldsymbol{X}_1 - \boldsymbol{X}_4 \quad \boldsymbol{X}_2 - \boldsymbol{X}_4 \quad \boldsymbol{X}_3 - \boldsymbol{X}_4]$.

Furthermore, the internal elastic forces on the vertices $(\boldsymbol{f}_1, \cdots, \boldsymbol{f}_4)$ can be computed though $[\boldsymbol{f}_1 \quad \boldsymbol{f}_2 \quad \boldsymbol{f}_3] = -\frac{1}{6}|\det \boldsymbol{D}_m|\boldsymbol{P}(\boldsymbol{F})\boldsymbol{D}_m^{-T}$ and $\boldsymbol{f}_4 = -\boldsymbol{f}_1 - \boldsymbol{f}_2 - \boldsymbol{f}_3$.

## L.3   Reduced-order Model

Since the initial condition is known for a given problem, we opt to construct a manifold-parameterization function for the displacement field $\boldsymbol{u}(\boldsymbol{X}, t) = \boldsymbol{\phi} - \boldsymbol{X}$. The deformation map can then be computed via $\boldsymbol{\phi} = \boldsymbol{u} + \boldsymbol{X}$. The hyperparameters of the manifold-parameterization function are $r = 2$ and $\beta = 20$.

Unlike previously discussed PDEs, the elastodynamics equation is second-order in time. Consequently, in addition to the function value of the manifold, we also need to infer the temporal derivative during latent space dynamics. We do so by employing the tangent space of the manifold,

$$\dot{\boldsymbol{\phi}}(\boldsymbol{x}, t_n) = \frac{\partial \boldsymbol{g}_{\theta_g}}{\partial \boldsymbol{q}} \dot{\boldsymbol{q}}_n. \tag{31}$$

, where $\dot{\boldsymbol{q}}_n = (\boldsymbol{q}_n - \boldsymbol{q}_{n-1})/\Delta t$ and $\dot{\boldsymbol{q}}_0 = \boldsymbol{0}$.

To compute the acceleration (and increment the velocity) of a vertex belonging to the integration samples set $\mathcal{M}$, we need to evaluate the internal force at this particular vertex. We can calculate the vertex force by accumulating the internal forces of tetrahedra incident on this particular vertex. Tetrahedron forces are computed using the formula described in the previous section. To attain optimal gradient accuracy (Appendix E.2), we opt to use finite element's linear basis function to compute the deformation gradient $\boldsymbol{F}$ of each tetrahedron.

To compute these internal forces and deformation gradients, we also need to obtain the position information (via the manifold-parameterization function) of all vertices in the one-ring neighborhoods of the integration samples. We refer to the set of these neighboring vertices as $\mathcal{N}$. Therefore, the total number of spatial samples in the reduced-order model now becomes $|\mathcal{M}| + |\mathcal{N}|$. Since one-ring neighbors only entail a small subset of the original degrees of freedom, we still offer significant spatial sample reduction from the full order model ($P$).

## L.4   Training and Testing Data

### L.4.1   Gravity-induced Impact

An initially-static rectangular-shaped deformable object ($P = 2,065$ vertices and $9,346$ tetrahedra) accelerates due to downward gravity. After impacting the collision objects underneath, the material undergoes intense deformation.

We generate training and testing data by sampling the shear modulus $\boldsymbol{\mu} = \mu \in \mathcal{D} = [60,000, 70,000] \subset \mathbb{R}$. The first lame parameter $\lambda$ is fixed to be zero, in which case the corotated linear elasticity constitutive law simplifies to the As-Rigid-As-Possible energy (Sorkine & Alexa, 2007). Specifically, we uniformly generate $4$ samples of the shear modulus for training data. We simulate $200$ time steps for each shear modulus. Therefore, a total of $800$ time steps are used for training. We then randomly generate $3$ samples of the shear modulus from $\mathcal{D}$. We also simulate $200$ time steps for these shear moduli and use them for testing. We use the same initial condition for training and testing.

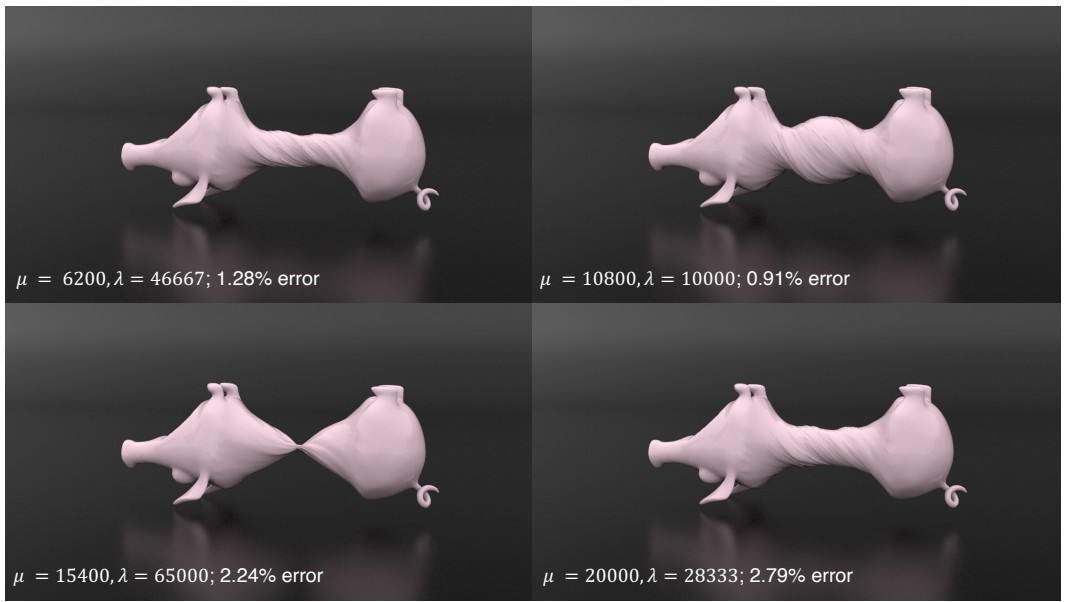

Figure 24: Our approach captures a wide range of elastic moduli with modest errors compared to the full-order ground truth.

### L.4.2    TORSION AND TENSION

We study another common solid mechanics application where the material undergoes torsion and tension. The left and the right boundaries kinematically move at constant translational and angular velocities of the same magnitude but in opposite directions.

We solve the parameterized family of problem where $\boldsymbol{\mu} = (\mu, \lambda) \in \mathcal{D} = [6, 200, 20, 000] \times [10, 000, 65, 000] \subset \mathbb{R}^2$. We obtain 9 training samples via uniform full-factorial sampling of the parameter space. We simulate 100 time steps for each training sample. We then use the Latin hypercube method to randomly generate another 4 testing samples. We also simulate 100 time steps for these testing samples. Training and testing data adopt the same initial condition.

### L.5    RESULTS

Figure 8 demonstrates a testing case ($\mu = 62500$, 2.68% error from the full-order ground truth) for the experiment from Appendix L.4.1. Since our method has a discretization-agnostic architecture, we can adapt the simulation resolution to improve efficiency. The adaptive simulation adopts a contact-based oracle that switches from the low-resolution mesh to the high-resolution mesh once a contact event is detected. Before the start of the simulation, we generated both the low-resolution and the high-resolution meshes using TetWild (Hu et al., 2018) with small and large ideal edge lengths.

After training on the experiment from Appendix L.4.2, our method captures a wide range of shearing and volume-preserving behaviors (see Figure 24).

Furthermore, to showcase CROM's ability to handle large deformations, we reproduce the dinosaur example by Shen et al. (2021). Figure 25 shows that CROM accurately captures the large deformations of the dinosaur. By contrast, under the same latent space dimension, POD has a limited expressivity and suffers from visual artifacts: the head and the tail's motions are constrained. This observation is consistent with the results reported by Shen et al. (2021) where nonlinear approaches outperform linear approaches. In terms of memory consumptions, CROM is orders-of-magnitude more efficient than these prior discretization-dependent ROM approaches (Barbič, 2012; Fulton et al., 2019; Shen et al., 2021).

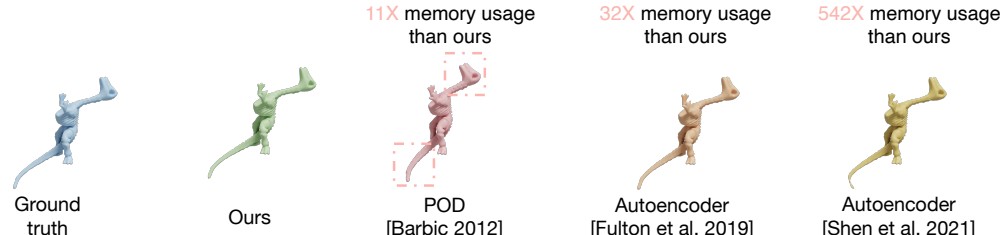

Figure 25: CROM agrees well with the ground truth simulation while consuming far less memory than prior discretization-dependent ROM approaches, including POD (Barbič, 2012) and neural-network-based autoencoder approaches (Fulton et al., 2019; Shen et al., 2021). The experimental setup is taken from Shen et al. (2021).

## M    COMPARISON WITH PRIOR APPROACHES

We compare our approach with prior reduced-order approaches that generate fast PDE solutions via latent space dynamics. In total, we compare with four baselines: (1) the classic widely-adopted proper orthogonal decomposition (POD) method (Berkooz et al., 1993; Holmes et al., 2012). Our POD implementation follows the PCA approach by Barbič & James (2005); Barbič (2012); (2) the neural-network-based autoencoder approach by Fulton et al. (2019); (3) the neural-network-based autoencoder approach by Shen et al. (2021); (4) the convolutional autoencoder (CAE) approach by Lee & Carlberg (2020). All these four baselines are *discretization-dependent*.

SIREN, PINN, or GNN are related to our work but are very different in nature. While their goal is to directly solve PDEs with neural network architectures, our goal is to leverage neural networks for dimension reduction. Therefore, we do not compare with them and focus on comparison with other dimension reduction methods.

POD approximates the discretized vector field with a linear basis $\boldsymbol{U}$,

$$\overrightarrow{\boldsymbol{f}}(t) = \left(\boldsymbol{f}(\boldsymbol{x}^1,t),\dots\boldsymbol{f}(\boldsymbol{x}^i,t),\dots\boldsymbol{f}(\boldsymbol{x}^P,t)\right)^T = \boldsymbol{U}\boldsymbol{q}(t), \tag{32}$$

where $\boldsymbol{U}$ is a matrix of size $Pd$ by $r$. $\boldsymbol{U}$ is most easily constructed via a singular value decomposition of the discretized training data (Sifakis & Barbic, 2012).

Autoencoders have the same input and output as POD. However, the linear basis is replaced with nonlinear neural networks,

$$\overrightarrow{\boldsymbol{f}}(t) = \left(\boldsymbol{f}(\boldsymbol{x}^1,t),\dots\boldsymbol{f}(\boldsymbol{x}^i,t),\dots\boldsymbol{f}(\boldsymbol{x}^P,t)\right)^T = \boldsymbol{h}(\boldsymbol{q}(t)), \tag{33}$$

where $\boldsymbol{h}$ is the decoder portion of the autoencoder structure. Fulton et al. (2019); Shen et al. (2021); Lee & Carlberg (2020) all follow this formulation but differ in their particular architecture choices.

We implement both POD and the autoencoder approaches in the same PyTorch framework as CROM. These approaches share exactly the same PDE time-stepping algorithm (step 2) during latent space dynamics. For POD and the autoencoder approach, the network inference step (step 1) and the network inversion step (step 3) employ the linear basis $\boldsymbol{U}$ and the nonlinear decoder function $\boldsymbol{h}$, respectively.

### M.1    SPEED

We measure the wall-clock computation time during the falling stage (before contact) of the gravity-induced impact experiment (see Appendix L.4.1 and Figure 8). In particular, since all model reduction methods (including our approach, POD, and the autoencoder approaches) share the same PDE time-stepping algorithm (step 2), we measure its wall-clock computation cost to ensure the generalizability of the comparison.

Since POD and the autoencoder approach are baked into a particular discretization, it does not allow for adaptive discretization. Therefore, the high-resolution mesh ($2,065$ vertices, $9,346$ tetrahedra) has to be used throughout the falling stage even though there is minimal deformation. Consequently, its computation cost is significantly higher than our approach, which employs a low-resolution mesh ($8$ vertices, $5$ tetrahedra) during the falling stage. Figure 26 shows that our approach is faster

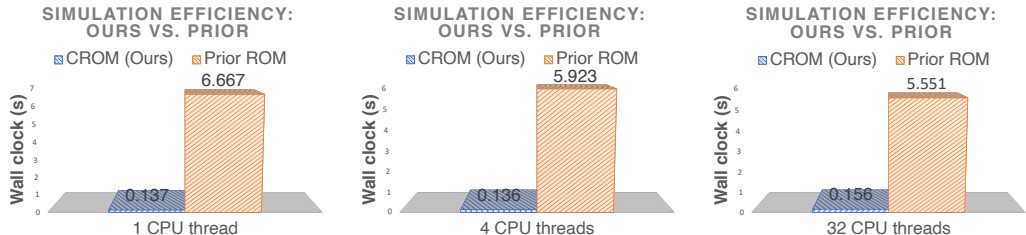

Figure 26: Wall clock time comparison with prior ROM approach. Our approach obtains considerable speedups across different computational resources.

than the prior ROM approach with different CPU thread counts. Our method's advantage is most obvious when there is extremely limited computational resource (1 CPU thread). Since this speedup comes from the discretization independent nature of CROM, we expect it to generalize to other prior discretization-dependent ROM approaches as well. However, for application where remeshing is unnecessary, we do not expect this kind of speedup.

Hyper-reduction techniques (An et al., 2008), such as the integration samples introduced in our approach, can also be employed to speed up the high-resolution mesh solution by reducing spatial samples. However, this is orthogonal to our adaptive discretization contribution, and the hyper-reduction techniques can be employed to further accelerate the low-resolution mesh solution as well.

## M.2 MEMORY

We compare the memory consumptions of our method, POD, and the autoencoder approaches with the same latent space dimension ($r$), both trained on data from the torsion and tension experiment (see Appendix L.4.2). For all implicit neural representation networks, we set the MLP width to be $\beta = 20$. Independent of latent space dimension ($r$), our network has a near-constant memory consumption (see Figure 10). By contrast, the memory consumptions of POD and the autoencoder approaches scale linearly with $r$. More importantly, since the output dimension of POD and the autoencoder is $Pd$, their memory consumptions scale linearly with the number of discretized positions $P$. By contrast, our approach's memory consumption is independent of the number of spatial samples. Consequently, in this large-scale example that features $P = 66,608$ vertices, we observe more than ten-fold advantages with our method across all latent space dimensions (see Figure 10).

In addition, we also compare our method with POD on the image processing experiment (see Appendix H). CROM uses an order-of-magnitude less memory than POD (see Figure 5c). In Figure 27, we also demonstrate CROM's memory advantage over the convolutional autoencoder approach by Lee & Carlberg (2020).

Even though the overall memory consumption of our approach is significantly lower, our approach's per-sample evaluation cost is higher. For any given discretized spatial sample, POD only requires a vector-matrix multiplication of a $d$ by $r$ matrix. By contrast, our approach needs to go through the entire MLP that employs $d\beta$ by $d\beta$ matrices. The autoencoder approaches share similar computation overheads.

## M.3 ACCURACY

Figure 10 demonstrates the accuracy of the networks studied in the previous memory section. Our method consistently offers orders-of-magnitude smaller training accuracies. This accuracy advantage is especially noticeable with small latent space dimensions (e.g., $r = 2$). Further increasing the latent space dimension leads to higher accuracies with POD and the autoencoder approaches but not our approach. As discussed in Appendix F, our approach's accuracy depends more saliently on the MLP width as opposed to the latent space dimension. Furthermore, unlike these discretization-dependent approaches, our continuous manifold-parameterization function also facilitates accurate handling of adaptive spatiotemporal data (Pan et al., 2023).

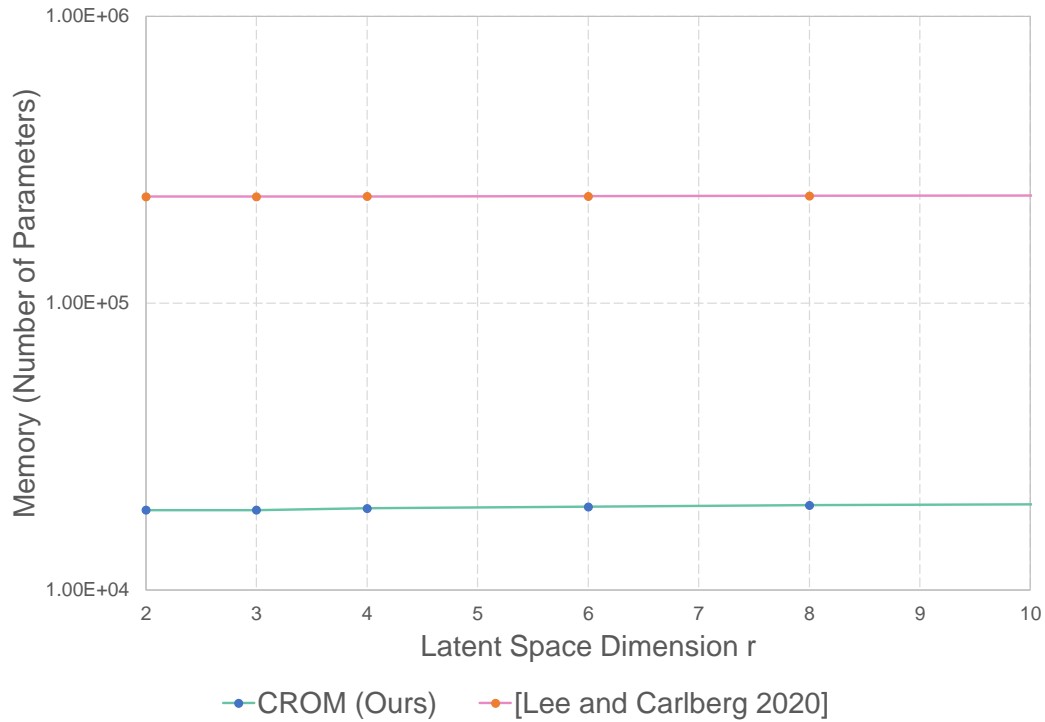

Figure 27: Memory comparison between our CROM and Lee & Carlberg (2020)

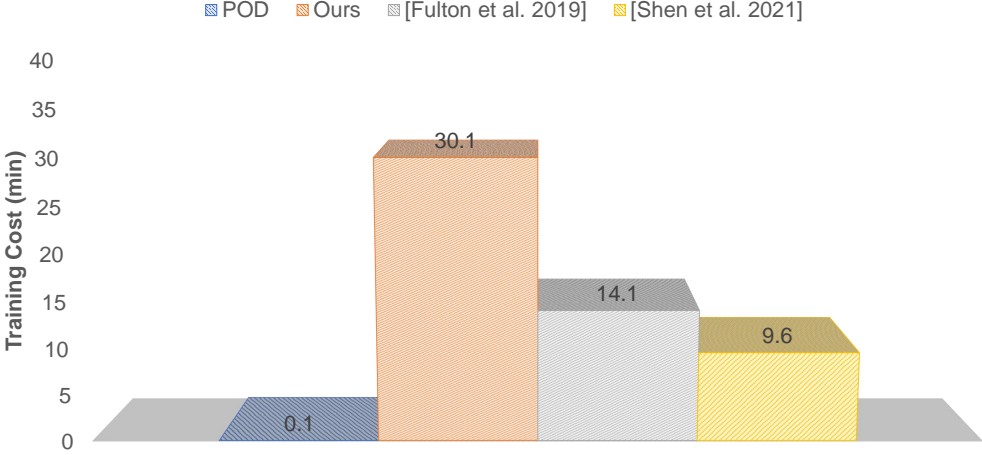

Figure 28: Training time for the elasticity example presented in Figure 25. POD (Barbič, 2012) uses significantly less time than neural-network-based methods, including (Fulton et al., 2019; Shen et al., 2021) and ours.

Furthermore, Figure 5 shows that our method is more accurate than POD on the image processing experiment, both visually and quantitatively. Figure 22 displays our accuracy advantage over CAE on Burgers' equations.

## M.4 TRAINING TIME

Figure 28 reports the offline training time for different ROM approaches. All neural-network-based methods, including ours, take longer time to train than the linear POD approach. We emphasize that the timing data is purely informative and calculated post-hod. The main target of our work is not to

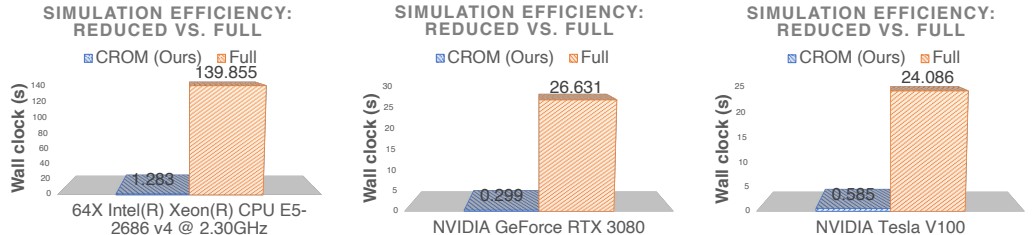

Figure 29: Wall clock time comparison with the full-order method. Our approach obtains significant speedups across different computing platforms.

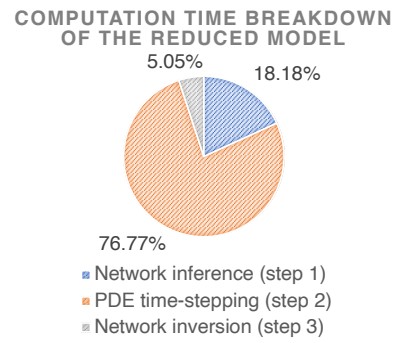

Figure 30: CROM timing breakdown. Neural network operations introduce minimal overhead. The computation bottleneck remains to be PDE time-stepping.

optimize training time but to achieve efficient latent space dynamics. Nevertheless, specifically for our implicit neural representation approach, future work may consider expediting training time via more advanced data structures and optimization (Müller et al., 2022; Liu et al., 2020; Martel et al., 2021; Takikawa et al., 2021).

## N    COMPARISON WITH THE FULL-ORDER MODEL

To gauge the practical performance of our approach over the (unreduced) full-order model, we measure the sheer wall-clock performances of our reduced-order model and the full-order model.

To facilitate fair comparison, we implement, optimize, and (fully) parallelize the full-order PDE solution and the reduced-order approach within the PyTorch framework (Paszke et al., 2019) without any other dependency. The full-order model and the reduced model share the same (parallelized) PDE solver code (step 2), and the only difference between them is the neural network evaluation in the reduced model. To encourage fair comparison, for operations involving neural networks (step 1 and step 3), we do not use customized SIMD vectorizations, CUDA kernels, or highly optimized inference libraries such as TensorRT (Vanholder, 2016), though these optimizations should lead to a further performance gain of our algorithm. In addition, having both models implemented in the same framework allows us to compare them under the same computing environments, avoiding biased comparison where the full-order model runs on a significantly limited computing resource (e.g., 8-core CPU) while the reduced-order model runs on a high-performance computing device (e.g., 5120-core GPU).

Figure 29 reports the wall clock time of our reduced-order method and the full-order method across different computing platforms. While our approach is significantly faster on all computing platforms, its strongest speedup is obtained on the CPUs. This is unsurprising since the computationally expensive full-order model benefits more from high-end data-center GPUs (e.g., V100) than the reduced-order model. In addition, because the neural network in the reduced-order model only employs a few 60 by 60 matrices ($d = 3, \beta = 20$), inference on CPU is also efficient. This opens doors for employing our models on limited computing platforms such as mobile and VR devices.

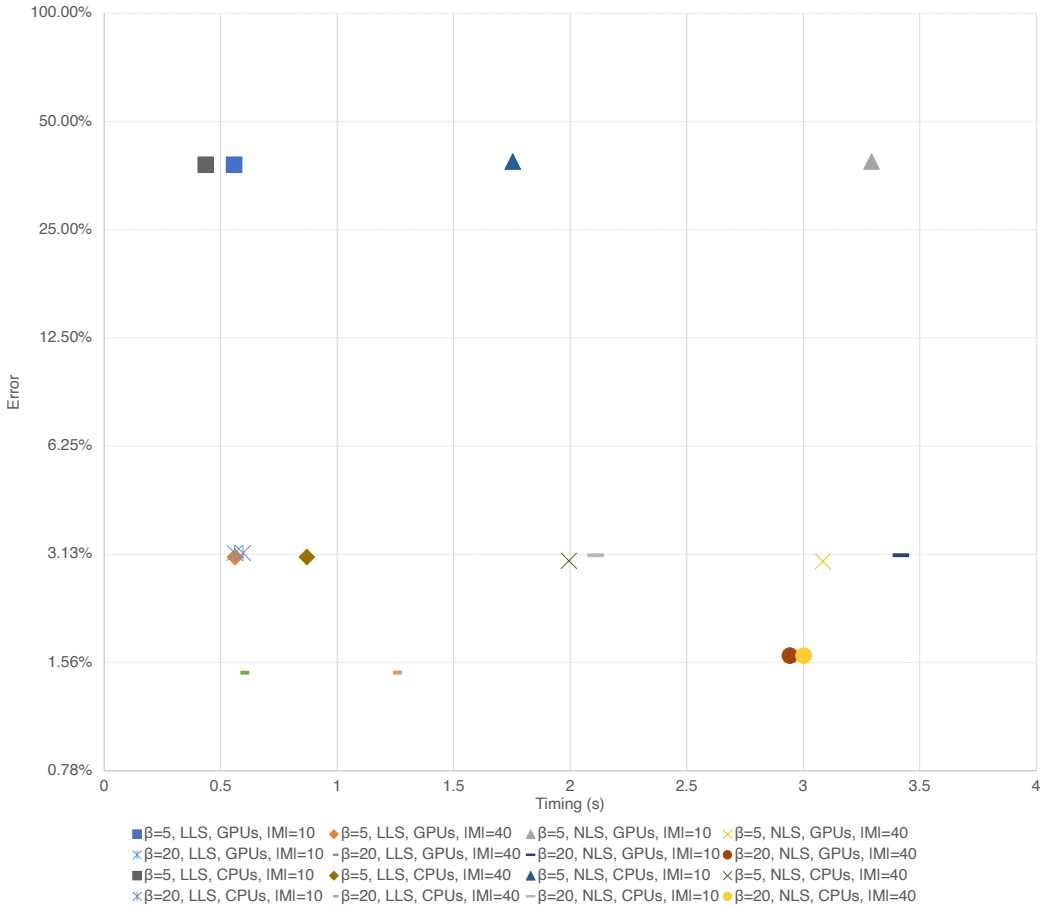

Figure 31: Accuracy vs. Speed: our approach under different setups. In comparison, the ground truth full-order takes 139.855s on CPUs and 24.086s on GPUs (see Figure 29). Our approach obtains significant speedups with less than 5% error under various settings.

Figure 30 further breaks down the computation time spent on each component of the reduced-order model. While the overhead of the neural network operations (step 1 and step 3) is non-negligible, the bottleneck of the algorithm remains to be PDE time-stepping (step 2). In particular, the network inversion cost is very low, thanks to the linearized inversion solver discussed in Appendix C.

Figure 31 reports the accuracy and the speed of our approach using different MLP layer widths ($\beta$), neural network inversion methods (nonlinear least squares (NLS) vs. linear least squares (LLS)), computing platforms (CPU vs. GPU), and integration sample sizes ($|\mathcal{M}|$). In general, we observe similar accuracies between NLS and LLS (e.g., [$\beta = 20$, LLS, GPUs, $|\mathcal{M}| = 40$] vs. [$\beta = 20$, NLS, GPUs, $|\mathcal{M}| = 40$]), while LLS is significantly faster. Increasing sample size lead to higher accuracies but also require longer computation time (e.g., [$\beta = 5$, NLS, CPUs, $|\mathcal{M}| = 10$] vs. [$\beta = 5$, NLS, CPUs, $|\mathcal{M}| = 40$]). Increasing the MLP layer width yields higher accuracies but also increases the computation time. In particular, the computation time is increased more on CPUs (e.g., [$\beta = 5$, LLS, CPUs, $|\mathcal{M}| = 40$] vs. [$\beta = 20$, LLS, CPUs, $|\mathcal{M}| = 40$]) than on GPUs.

## O  LATENT-SPACE TRAJECTORY

A key element of CROM is latent space dynamics, in which the latent space vector evolves nonlinearly over time under explicit PDE constraints.

Figure 32 demonstrates a *nonlinear* latent space trajectory from the image smoothing experiment. Guided by the PDE, the latent space vector traverses through the latent space and obtains the desirable smoothing effects in the image (see Figure 33).

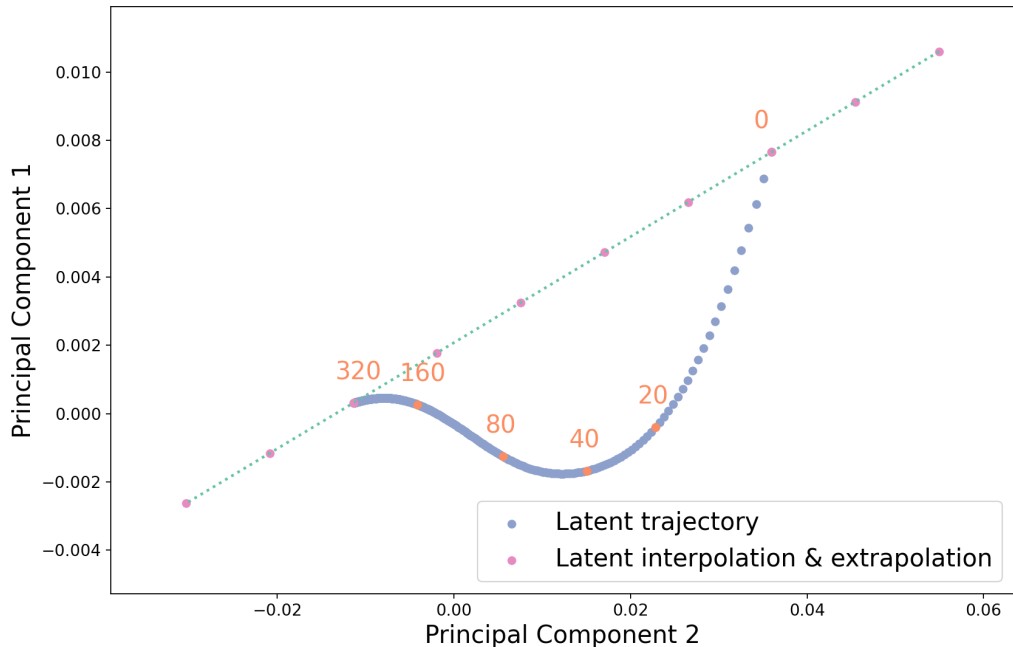

Figure 32: Latent-space trajectory. We visualize the first two principal components of the latent space vector. The red numbers displayed on top of the trajectory are time step numbers. The simulation starts from 0 and ends at 320. The corresponding images of these time steps are shown in Figure 33. Overall, we observe a smooth nonlinear trajectory over time. In addition, we also linearly interpolate and extrapolate inside the latent space. The corresponding images are displayed in Figure 34.

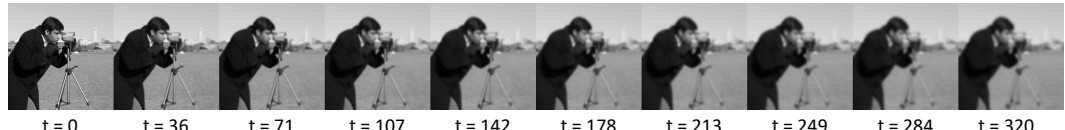

Figure 33: Reduce-order simulation of image smoothing at at different time steps $t$. The corresponding latent space trajectory is visualized in Figure 32.

After the simulation is finished, as a postprocessing experiment, we *linearly* interpolate and extrapolate inside the latent space (see Figure 32). The interpolation end points are the latent vector for the unmodified image and the latent vector for the final image determined by the latent space dynamics. Figure 34 demonstrates the corresponding smooth transitions in the image space.

## P   STABILITY ANALYSIS

Figure 35 systematically studies the stability of our method. CROM exhibits much larger stable time step sizes than its unreduced counterpart. This is consistent with the theoretical analysis reported in the literature (Bach et al., 2018).

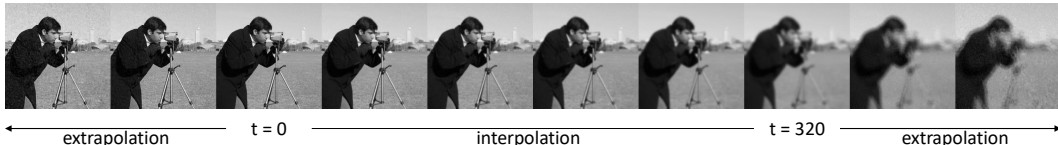

Figure 34: Linear interpolation and extrapolation of the latent space vector. We observe smooth transitions. The corresponding latent space vector values are shown in Figure 32.

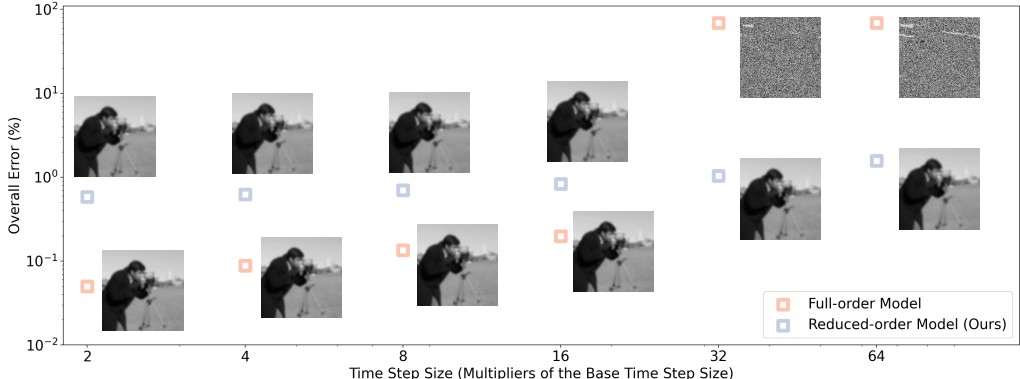

Figure 35: Stability analysis. We systematically increase the time step size ($\Delta t$). Our method remains stable in a wide range of time step size and maintains an error close to $1\%$. By contrast, the full-order model (that was used for training data generation) goes unstable as the time step increases and suffers from major visual artifacts.

## Q  LATENT-SPACE DYNAMICS PSEUDOCODE

---

**Algorithm 3:** Latent-space Dynamics

---

**Input:** Latent space vector $\boldsymbol{q}_n$
**Output:** Next-time-step latent space vector $\boldsymbol{q}_{n+1}$

1   Network inference: gather full-space information (Algorithm 4).
2   PDE time-stepping with boundary conditions: update full-space information via differential equations, e.g., the elastodynamics equation (more details available from Appendix G to Appendix L).
3   Network inversion: optimally project the updated full-space information onto the low-dimensional embedding (Algorithm 5).

---

**Algorithm 4:** Network inference: gather full-space information

---

**Input:** Latent space vector $\boldsymbol{q}_n$
**Output:** Full space information $\boldsymbol{f}_n, \nabla \boldsymbol{f}_n, \dot{\boldsymbol{f}}_n$

1   **for** $\boldsymbol{y} \in \mathcal{M}$ **do**
2     Compute the vector field value: $\boldsymbol{f}(\boldsymbol{y}, t_n) = \boldsymbol{g}_{\theta_g}(\boldsymbol{y}, \boldsymbol{q}_n)$.
3     Compute the temporal gradient either by differentiating the network $\dot{\boldsymbol{f}}(\boldsymbol{y}, t_n) = \frac{\partial \boldsymbol{g}_{\theta_g}}{\partial \boldsymbol{q}} \dot{\boldsymbol{q}}_n$,
     where $\dot{\boldsymbol{q}}_n = (\boldsymbol{q}_n - \boldsymbol{q}_{n-1})/\Delta t$, or by numerical approximation (Appendix E).
4     Compute the spatial gradient either by differentiating the network, $\nabla \boldsymbol{f}(\boldsymbol{y}, t_n) = \nabla_{\boldsymbol{y}} \boldsymbol{g}_{\theta_g}$, or
     by numerical approximation (Appendix E).
5   **end**

---

**Algorithm 5:** Network inversion: optimally project the updated full-space information onto the low-dimensional embedding

---

**Input:** Updated full space information $\boldsymbol{f}_{n+1}$
**Output:** Updated latent space vector $\boldsymbol{q}_{n+1}$

1   Find the latent space vector $\boldsymbol{q}_{n+1}$ that best matches the evolved configuration $\boldsymbol{f}_{n+1}$ by solving the least-squares:

$$\min_{\boldsymbol{q}_{n+1} \in \mathbb{R}^r} \sum_{\boldsymbol{y} \in \mathcal{M}} \|\boldsymbol{g}_{\theta_g}(\boldsymbol{y}, \boldsymbol{q}_{n+1}) - \boldsymbol{f}(\boldsymbol{y}, t_{n+1})\|_2^2 \,.$$

---

| PDE | Full-order degrees of freedom | Latent space dimension | Dimension reduction | Spatial sample count | Spatial sample reduction |
|---|---|---|---|---|---|
| 1D Thermo | 501 | 16 | $31\times$ | 22 | $23\times$ |
| 2D Image | 65,536 | 16 | $4,096\times$ | 63 | $1,040\times$ |
| 1D Advection | 100 | 1 | $100\times$ | n/a | n/a |
| 1D Burger | 256 | 2 | $128\times$ | n/a | n/a |
| 2D fluid | 512 | 6 | $86\times$ | n/a | n/a |
| 3D solid impact | 6,195 | 2 | $3,098\times$ | n/a | n/a |
| 3D solid piggy | 199,824 | 2 | $99,912\times$ | 40 | $1,665\times$ |
| 3D solid dinosaur | 11,565 | 1 | $11,565\times$ | n/a | n/a |

Table 1: Dimension reduction statistics of our reduced-order model

# R    REDUCED-ORDER MODEL'S STATISTICS

CROM obtains considerable dimension reductions across different PDEs (Table 1).

