# OpenReview forum: "CROM: Continuous Reduced-Order Modeling of PDEs Using Implicit Neural Representations"
_ICLR.cc/2023/Conference — ICLR 2023 notable top 25%_

### Official Review · Reviewer_7uQY · 2022-10-24

**Confidence:** 4
**Correctness:** 4
**Technical Novelty And Significance:** 3
**Empirical Novelty And Significance:** 3
**Recommendation:** 8

**Clarity, Quality, Novelty And Reproducibility:**

Very clear motivations, technical exposition and experimental presentation. Fully reproducible, partially thanks to an extensive (but not too verbose) supplementary material.

**Strength And Weaknesses:**

Using neural networks to reduce the dimensionality of physics phenomenon in a learned and non linear but smooth manner is a very good motivation, and as noted by the authors it had already been explored. Here it is also combined (for the first time) with an IMPLICIT representation, which allows for quantization-free modelling.
The latter point has two benefits: the pipeline requires less memory, and can easily adapt to any resolution by simply querying the network more or less densely.

The experimental section is very rich, displaying the applicability of the method for many applications (image blurring, heat equation, advection, elastodynamic deformation, …). Comparing network outputs to a full fledged PDE solver (slow) and other network based solvers quantitatively and qualitatively favors the proposed pipeline: it is almost on par with the full PDE solver, and largely more accurate than other network based approaches.
A runtime and memory analysis also demontrates that the proposed approach is much more efficient once trained.

My main doubt is that such a pipeline could only (locally) interpolate between seen behaviors from the training set. How does this prevent applicability? How much care must be taken in crafting a training set that spans a sufficiently large spectrum of possible dynamics?

Another concern is about runtime comparison with full order models: for a fair comparison, maybe the training set generation and network training time would have to be accounted for in some way. Indeed, the trained approach can be seen as a way to smartly “memorize” simulation results to do much faster predictions. But this has an upfront computational cost, which only gets amortized when the resulting network is used for many different simulations.

**Summary Of The Paper:**

This paper proposes to use implicit neural representations to model and efficiently solve PDE dynamics. This allows to solve dynamics in the latent space instead of the whole parameter space (lower dimension), without discretizing it. The latent space and network are trained on a range of sample PDEs for a given task, and employed to solve PDEs with different parameters.
Solving the temporal evolution is still done with an explicit PDE constraint at each time step, providing updates of latent space parameters, instead of the explicit parameters which have much higher dimensions.

**Summary Of The Review:**

Clear motivation, elegant method and strong experimental section.

---

> ### Author Response · Authors · 2022-11-14
> **Response to Reviewer 7uQY**
>
> - Q: How to design training set to capture large spectrum of possible dynamics?
>
> - A:
>
>   - There are two camps of work for designing training sets for ROM methods.
>   - The first line of work follows the design of experiments techniques. Given a parameter range of interest D, we can generate training simulation data by drawing samples from D, e.g., uniform sampling, latin hypercube \[3]. Our method follow this line of work. Alternatively, users can also generate training data via crowed sourcing without explicit define the range of D, e.g., possible deformations by users in a Virtual Reality setting \[4].
>   - Another lines of work takes an active learning approach (also known as greedy/adaptive sampling approach) \[5]. In a nutshell, whenever a parameter of interest entails a larger error on the current trained model, this approach adds it to the training set. We believe this is an exciting future direction. Other implicit neural reprsentation methods (e.g., NeRF and DeepSDF) can also improve their generalizability via these active learning methods.
>
> - Q: “Fair comparison w/ the full order model should include training time”
>
> - A:
>
>   - We have since added the training time for reference (see appendix M.4). We emphasize these numbers are recorded post-hoc and can be further opitmized using SOTA implicit neural representation training techniques \[1]. As the reviewer pointed out, the training time “gets amortized when the resulting network is used for many different simulations.” One of the target applications of our method is real-time physics for virtual reality \[2]. In this case, a trained model would be deployed on the cloud and accessed by millions of users. As such, the training time can be considered negligible.
>
> We appreciate reviewer’s comments on training data design and training time comparision!  Please consider raising the rating after taking ours responses into consideration. We look forward to incorporating any suggestions on improving the quality of the manuscript.
>
> \[1] Müller+ Instant neural graphics primitives with a multiresolution hash encoding, SIGGRAPH 2022
>
> \[2] Savva+ Habitat: A platform for embodied ai research, ICCV 2019
>
> \[3] Carlberg+ The GNAT method for nonlinear model reduction: effective implementation and application to computational fluid dynamics and turbulent flows. Journal of Computational Physics, 2013
>
> \[4] Barbic+ Real-Time Subspace Integration for St.Venant-Kirchhoff Deformable Models. SIGGRAPH 2005
>
> \[5] Grepl+ A posteriori error bounds for reduced-basis approximations of parametrized parabolic partial differential equations, ESAIM: Mathematical Modelling and Numerical Analysis 2005
>
> \[6] Mildenhall+ Nerf: Representing scenes as neural radiance fields for view synthesis, Communications of the ACM 2022
>
> \[7] Park+ Deepsdf: Learning continuous signed distance functions for shape representation, CVPR 2019

---

### Official Review · Reviewer_XNfR · 2022-10-24

**Confidence:** 3
**Correctness:** 3
**Technical Novelty And Significance:** 3
**Empirical Novelty And Significance:** 3
**Recommendation:** 6

**Clarity, Quality, Novelty And Reproducibility:**

## Clarity
Even if the paper is clearly drafted, I have some questions about some of the steps in the method.

### Step 1: Network Inference
It is not clear to me how you compute $\dot{q_n}$ ?

### Step 2: PDE Time-stepping
Again about time derivative, how do you compute $\dot{f_{n+1}}$ ?

### Spatial Sample reduction

You say that the well-posedness depends strongly on the selection of the points where you solve the PDE.
I have several concerns about this selection.

You arguee that eq(7) is well-posed if $r \leq d|M|$, I am not sure why?

About the integration sample, I have two main concerns.
First, in introduction of part 4, you say that integration samples are chosen at the user’s discretion, which is in fact not true as you give an algorithm (introduced in Appendix B). I think you should give more insights in the main paper about this greedy algorithm and how you select the mesh you are solving the PDE on. Notably, how do you fix $M$?

Second, you say in Appendix B that Algo 1 gives a pre-computation, what do you mean by that?

Finally, adaptative resolution seems like a great benefit in your paper. However, you remain quite elusive on how you implement it, could give some insights ?

### Experiments

To what models are you comparing to ?
You refer to two papers, but show only one image of POD in Fig 5.

## Vocabulary

You’re talking about a low-dimensional manifold $g_{\theta_g}$, which sounds weird to me. I would rather say that the low-dimensional manifold is your latent space $q$.

You are talking about “latent space dynamics” in both Fig 1 and Part 5 (where you say that you "validate the latent space dynamics").
However, you don't really learn this dynamics, as you consider known the PDE and you use it to predict the next $q_{t+1}$. This is somewhat misleading.


**Strength And Weaknesses:**

On overall, the paper is clear, well-written and easy to follow. The method has many advantages compared to traditional ROM approaches, among which applicability to any discretization (which appears as a great deal of current research in dynamical systems) and reduced memory footprint and consumption.
However, some steps in the method are not mentioned in the main paper, only in the appendix, which hinders the overall comprehension of the implementation, as well as its reproducibility. Also, some statements are misleading (developed hereafter).

**Summary Of The Paper:**

The paper proposes to use a reduced order model (ROM) to accelerate PDE resolution.
They differ from prior ROM approaches by directly modeling the continuous function of interest, and not an already discretized function.
To approximate the continuous PDE solution, CROM relies on a latent space and use implicit neural representations as decoder, which enable them to address any discretization. The PDE is solved in a lower dimensional space, enabling them lower memory consumption. Experiments are conducted on several PDE depicting various discretizations, and comparisons with several baselines are performed.

**Summary Of The Review:**

The paper combines various recent methods (ROM, PINN, INR) to tackle an interesting problem, namely the acceleration in the resolution of PDEs. Some minor shortcuts hinder the comprehension of the method. To add clarifications on the first and last step of the implementation would enhance the overall quality.

---

> ### Author Response · Authors · 2022-11-14
> **Response to Reviewer XNfR (Part I)**
>
> We appreciate the detailed and insightful response. We have since done a pass over the paper to fix any part that is overly concise. Here we summarize the major updates:
>
> - Q: add clarifications on the first and last step of the implementation would enhance the overall quality.
> - A: We clarify step 1 and 3 of the latent space dynamics by adding the corresponding pseudocodes (see appendix Q).
> - Q: reproducibility
> - A: In addition to textual changes, here is the link to the source code: <https://www.dropbox.com/s/dj8lwk4mp34mk5i/crom_source_code.zip?dl=0>. We will also open-source the code in the near future.
> - Q: how to compute the qndot used in section 4.1
> - A: We assume a first-order approximation. Specifically, qndot =  (qn+1 - qn) / dt. We also assume the initial latent velocity is 0. In all examples presented in this paper, qndot is only involved in the elastodynamics equation, which is second-order in time. Further details are provided in appendix L.3.
> - Q: how to compute the fdotn+1 used in section 4.2
> - A: fdotn+1 is computed from eqn 5, which depends on the PDE of interest. For example, eqn 17 and eqn 22 in appendix details how it is computed for the thermondyanics and the image processing eamples.
> - Q: why is eq(7) well-posed if r&lt;= d|M|
> - A: eq(7) is a least-sqaures problem \[7]. A necessary condition for the well-posedness is that the number of equations (d|M|) is more than the number of variables(r). However, this is not a sufficient condition. \[7] demonstrates convergence of Gauss-Newton under the condition Jacobians J(x) have their singular values uniformly bounded away from zero. We have since clarified in the text that this is only a necessary condition.
> - Q: how to choose the integration samples?
> - A: (1) Yes, in principal, the users can choose the integration samples however they like; (2) In this work, we provide a recommended heuristic for choosing these integration samples, i.e., the greedy algorithm; (3) This strategy iteratively adds spatial samples to the sampling set until the target residual is met. In this case, the user decides the target accuracy, not the sample set size. Alternatively, the user can terminate the algorithm early once it reaches a chosen sample set size (see section 4.4).
> - Q: is the sampling heuristic a pre-computation?
> - A: It is a pre-computation in the sense that is done before solving PDEs via the proposed latent space dynamics (section 4.2). That is, after training the low-dimensional manifold (section 4.1), we choose the spatial samples via this sampling heuristic. Afterwards, we may solve the latent space dynamics on using these samples for any testing problem setups.
> - Q: adaptive resolution details
> - A: The adaptive simulation adopts a contact-based oracle that switches from the low-resolution mesh to the high-resolution mesh once a contact event is detected. Before the start of the simulation, we generated both the low-resolution and the high-resolution meshes using TetWild \[1] with small and large ideal edge lengths (see appendix L.5 first paragraph)
> - Q: refer to two sources for POD, but only show one image of POD in Fig 5.
> - A: It is common to cite these two sources when referring to the POD method. See chapter 12 of the commonly adopted textbook Data-Driven Science and Engineering \[2]. The first source \[3] is the original paper while the second \[4] is the textbook written by the same authors.
> - Q: why is g called the low-dimensional manifold
> - A: Yes, thanks for pointing this out! A more accurate term for g is the manifold-parameterization function \[8]. Here the PDE solution lies on the codomain of g, which defines a manifold from the extrinsic view. The latent space is the domain of g, a low-dimensional embedding of the manifold. We have updated the manuscript.
> - Q: The name “latent space dynamics” is misleading. The dynamics are not learned.
> - A: The phrase “latent space dynamics” itself means evolving the latent space vector over time. Within the ROM community, it has been used for both the intrusive \[5] and nonintrusive \[6] techniques. Intrusive techniques, like ours, drive the dynamics in the latent space via the original PDE which involves NO learning; Nonintrusive techniques additionally learn a time-stepping operator and use it to drive the dynamics.
>
> We appreciate reviewer’s valuable suggestions on improving the clarity and vocabulary of the paper! Please consider raising the rating after taking these revisions into consideration. We look forward to incorporating any suggestions on improving the quality of the manuscript.

---

> > ### Author Response · Authors · 2022-11-14
> > **Response to Reviewer XNfR (Part II)**
> >
> > [1] Hu+ Tetrahedral meshing in the wild, SIGGRAPH 2018
> >
> > [2] Brunton and Kutz, Data-Driven Science and Engineering: Machine Learning, Dynamical Systems, and Control
> >
> > [3] Berkooz+ The proper orthogonal decomposition in the analysis of turbulent flows, Annual review of fluid mechanics 1993
> >
> > [4] Holmes+ Turbulence, coherent structures, dynamical systems and symmetry, Cambridge university press 2012
> >
> > [5]Fulton+  Latent-space Dynamics for Reduced Deformable Simulation, EUROGRAPHICS 2019
> >
> > [6] Maulik+ Time-series learning of latent-space dynamics for reduced-order model closure, Physica D: Nonlinear Phenomena 2020
> >
> > [7] Nocedal and Wright, Numerical optimization 1999
> >
> > [8] Lee+ Model reduction of dynamical systems on nonlinear manifolds using deep convolutional autoencoders, JCP 2020

---

### Official Review · Reviewer_yRQN · 2022-10-25

**Confidence:** 4
**Correctness:** 4
**Technical Novelty And Significance:** 3
**Empirical Novelty And Significance:** 3
**Recommendation:** 8

**Clarity, Quality, Novelty And Reproducibility:**

* Clarity: This paper is well-written.
* Novelty & Quality: The paper proposes an interesting construction bridging two emerging topics: a hybrid/neural PDE Solver and INR. It achieves significant gains in accuracy and efficiency.
* Reproducibility: The pipeline is complicated and requires many details. Without more implementation details or code, the main paper itself is not sufficient to reproduce this work. I checked the supplement, but it does not include code.

**Strength And Weaknesses:**

Strengths:
* The paper is well-written and explains complex concepts and the pipeline very clearly. Considering the numerous mathematical concepts, this paper is impressively concise. Especially the breakdown of the proposed method through Section 4.1 to 4.3 with Figure 3 makes this paper accessible to a general audience, I think.
* The combination of Neural PDE Solver / ROM and Implicit Neural Networks is an interesting and promising direction, although implicit neural networks are usually nothing but MLPs.
* The proposed method achieved significant performance gain as well as speed-up.
* The proposed method has the flexibility of INR that can be applied to any scale/resolution.

Weaknesses:
* The authors did not compare the training time. The key to high-quality solutions here is the quality of manifolds/basis functions. Also, the learned basis functions could be obtained by modulation using mu or features from the encoder e_{\theta_e}. It usually requires a non-trivial amount of data and training time. The efficiency of training needs to be compared. Also, the model sizes need to be compared for a fair comparison with the baseline method.

Questions:
1. How was g modulated by mu? In the literature, INR is often modulated by Hypernetworks that output the model parameters for another neural network.
2. Training of the encoder is key to accurate solutions. How do you generate the training data? Is the method sensitive to the distribution of training data?


**Summary Of The Paper:**

This paper proposes a new way to solve PDEs using implicit neural representations (INRs) combined with the discrete time discretization/integration and neural network inversion. By leveraging lower dimensional manifolds or learned basis functions via INRs, the proposed method achieved efficient numerical schemes to temporally evolve solutions on a lower dimensional manifold. Also, the high-quality lower dimensional manifold that can be modulated by problem parameters allows more accurate solutions. The experiments demonstrate that the proposed method is efficient and more accurate than a baseline method ROM.


**Summary Of The Review:**

Overall, this paper is well-written and studies a new combination to solve PDE more efficiently and accurately compared to other alternatives. By learning basis functions/manifolds with INR, the proposed method achieved impressive speed-ups and memory efficiency on top of higher approximate quality. This is a promising result and will inspire many other researches.

---

> ### Author Response · Authors · 2022-11-14
> **Response to Reviewer yRQN**
>
> - Q: Training time should be included.
>
> - A: We have since added the training time for both our approach and the baseline methods. (see appendix M.4)
>
> - Q: model sizes need to be fairly compared
>
> - A: All models are compared at the same latent space dimension (see section 4, first paragraph)
>
> - Q: Reproducibility
>
> - A: This link contains the source code: <https://www.dropbox.com/s/dj8lwk4mp34mk5i/crom_source_code.zip?dl=0>. We will also open-source the code in the near future.
>
> - Q: How does \\mu influence g?
>
> - A:
>
>   - In our work, \\mu influence g in two ways. (1) we generate training data by varying \\mu. As such, g is trained to reconstruct PDE solutions of various \\mu; (2) After training, during online latent space dynamics, \\mu is explicitly baked in during the PDE timestepping stage, i.e., section 4.2, eqn 5 and 6. Taking the simplest example of the heat equation (Appendix G). \\mu is the diffuse coefficient \\nu. It explicit participate the PDE timestepping, describing how much heat is diffused per time step (see appendix G.2, eqn 17).
>   - Unlike other implicit neural representations’ works, we have a g with a fixed weights for all the \\mu’s participating training and testing.
>
> - Q: how do you generate training data?
>
> - A:
>
>   - There are two camps of work for obtaining training data for ROM methods.
>   - The first line of work follows the design of experiments techniques. Given a parameter range of interest D, we can generate training simulation data by drawing samples from D, e.g., uniform sampling, latin hypercube \[1]. Our method follow this line of work. Sampling details are provided in the appendix along with each PDE (e.g., See appendix G.4).
>   - Another lines of work takes an active learning approach (also known as greedy/adaptive sampling approach) \[2]. In a nutshell, whenever a parameter of interest entails a larger error on the current trained model, this approach adds it to the training set.
>
> We appreciate reviewer’s suggestion of including training time in the comparision! Please consider raising the rating after taking this new data and other responses into consideration. We look forward to incorporating any suggestions on improving the quality of the manuscript.
>
> \[1] Carlberg+ The GNAT method for nonlinear model reduction: effective implementation and application to computational fluid dynamics and turbulent flows. Journal of Computational Physics, 2013
>
> \[2] Grepl+ A posteriori error bounds for reduced-basis approximations of parametrized parabolic partial differential equations, ESAIM: Mathematical Modelling and Numerical Analysis 2005

---

### Official Review · Reviewer_YNJM · 2022-11-03

**Confidence:** 4
**Correctness:** 3
**Technical Novelty And Significance:** 3
**Empirical Novelty And Significance:** 4
**Recommendation:** 8

**Clarity, Quality, Novelty And Reproducibility:**

The paper is clear and well written. The method is new and the reproducibility of the work is ensured by an exaustive Appendix section.

**Strength And Weaknesses:**

The approach can be seen as ROM-PINNs method, as also asserted by the authors. Strength And Weaknesses of the approach are then related to the PINNs approach plus the benefits introduced by the reduced representation of the manifold. The proposed approach can effectively accelerate the numerical calculation of PDE without being agnostic to the physics. The same tool can also be used to dinamically remesh the high dimensionla solution. Nevertheless, despite "The ROM community has demonstrated that these three steps can yield strong long-time stability even on stiff and chaotic dynamical systems" (cit.) the PINNs are well known to fail with chaotic systems. Therefore, it would have been interesting to understand whether or not model reduction is of benefit to the solution of chaotic dynamical systems also when a PINNs-like criteria is used to evolve the solution in time.

Despite the evolution of the solution in time is mesh-free, the whole method is not. In the Network Inference Step the latent vector at time zero is assumed to be given. Since the latent vector is the result of an encoder network, a discretized vector field (or at least a point-cloud) is always necessary. For PINNs a discretized vector field is also necessary in the training stage to enforce the initial condition, but this is not needed in inference.

The main limitation of the approach concerns the projection of the high dimensional solution into a low dimensional space. Authors claim that the low dimensional manifold is smooth. The smoothness is proven empirically for a rather simple test case but is not enforced in the training stage. Any kind of regularization is introduced for the latent state. This might be a major issue when dealing with physical systems that can undergo transitions of bifurcations. I suspect that the experiments in the paper are rather sipmle to experience such issue.

**Summary Of The Paper:**

The CROM approach proposes to solve the driving high dimensional PDE in a low-dimensional manifold, without having to discretize the continuous high-dimensional vector field. The discretized high dimensional spatio-temporal solution is projected in the low dimensional manifold through an encoder. The encoder produces a latent representation of the continuous vector field. A positional-dependent decoder is then employed to reconstruct the continuous-vector field. In order to evolve the PDE in time few saples of the continuous vector field are required to minimize the residual of the driving PDE with a PINNs-like loss. The limited number of spatial samples y and the low dimension of the latent representation allow to boost the solution of the PDE with a considerable reduction of computational resources (e.g. time and memory) without degrading the accuracy of the solution.

Several technical solutions are also proposed to effectively solve the dynamics of latent space (e.g. greedy sapling of the spatial points, numerical approximation of the gradients, solution of the least squares problem for the network-inversion).



**Summary Of The Review:**

The proposed approach is novel and interesting. The paper might have a coniderable impact in machine learning and numerical simulation community. For this reason I recommand acceptance of the paper. I suggest to the authors to be clearer on the limitations of the approach (i.e. discretized field needed to initialize the latent state, no guaranties on the smoothness of the latent space at least in the proposed version).

---

> ### Author Response · Authors · 2022-11-14
> **Response to Reviewer YNJM**
>
> - Q: application to chaotic system
>
> - A:
>
>   - The reduced-order modeling (ROM) / model reduction community has long been investigating methods handling chaotic system, including (1) structure preservation \[1], (2) discrete-time optimization \[2], (3) space-time projection \[3], and (4) replacing the ill-conditioned initial value problems with the well-conditioned ones \[4]
>
>   - Our approach is a ROM approach by nature. It is fully compatible with the four ‘fixes’ above.
>
>   - While our approach shares similar network architecture as PINN, we differ in significant ways.
>
>     - Training time:
>
>       - PINN enforces both a function loss and a PDE loss. For chaotic system, satisfying the PDE loss term is particularly challenging due to extreme nonlinearity.
>       - Our approach only enforces a function loss and effectively learns "only" the spatial / kinematics representation. At training time, we do not enforce any PDE loss term, which is significantly easier to train than PINN. This kind of kinematic loss term is a standard practice of the ROM community.
>
>     - Inference/Test time:
>
>       - PINN obtains solutions by a forward evaluation of the trained network.
>       - Our approach evolves the latent space according to a PDE constraint. Essentially, instead of enforcing PDE constraints during training time (which PINN does and making the training more challenging), our approach enforces PDE constraints during test time. This approach is consistent with all projection-based ROM methods, which best describe our approach.
>
>   - To demonstrate this point, we have added the Karman vortex street experiment. See <https://www.dropbox.com/s/bk019kywmx7nw9d/karman_video.mp4?dl=0>. More details are listed in Appendix K.5. Karman vortex street is particularly challenging as the flow bifurcates from a steady state to the periodic vortex shedding regime. To capture such an effect, the latent space dynamics needs to be robust under extreme nonlinearity. PINN is well-known to fail in this case \[9]. Our approach faithfully captures the initial steady state, bifurcation, and the vortex shedding regime.
>
> - Q: The method is NOT entirely mesh free. A discretized vector field is used to get the initial latent space.
>
> - A: We agree with the reviewer and have sinced clarified this in the manuscript.
>
> - Q: The low-dimensional manifold does not have smoothness guarantee.
>
> - A: We agree with the reviewer that for cases involving bifurcations \[8], the smoothness of the latent space would be particularly important. We can achieve this by introducing smoothness regularization term, such as such as employing a Jacobian penalty term \[5], adding a local isometry term \[6], and enforcing Lipschitz constraints \[7] (see appendix C.1). That said, we also demonstrate that under the current framework, we are able to capture the bifurcation seen in the Karman vortex street experiment. Our hypothesis is that the current encoder structure amounts to some level of smoothness regularization to the latent space.
>
> We appreciate reviewer’s suggestion of chaotic systems and comment regarding the relationship between our work and PINN. Please consider raising the rating after taking the Karman vortex experiment into consideration. We look forward to incorporating any suggestions on improving the quality of the manuscript.
>
> \[1] Carlberg+ Conservative model reduction for finite-volume models. Journal of Computational Physics, 2018
>
> \[2] Carlberg+ The GNAT method for nonlinear model reduction: effective implementation and application to computational fluid dynamics and turbulent flows. Journal of Computational Physics, 2013
>
> \[3] Parish+ Windowed least-squares model reduction for dynamical systems. Journal of Computational Physics, 2021
>
> \[4] Wang+ Least squares shadowing sensitivity analysis of chaotic limit cycle oscillations. Journal of Computational Physics 2014
>
> \[5] Chen+ Model Reduction for the Material Point Method via an Implicit Neural Representation of the Deformation map, arXiv, 2021
>
> \[6] Du+ Learning signal-agnostic manifolds of neural fields, NeurIPS 2021
>
> \[7] Liu+ Learning smooth neural functions via lipschitz regularization, SIGGRAPH 2022
>
> \[8] Brush+ Buckling of bars, plates, and shells, 1975
>
> \[9] Chuang+, Experience report of physics-informed neural networks in fluid simulations: pitfalls and frustration, arXiv 2022

---

### Official Review · Reviewer_NfXC · 2022-11-03

**Confidence:** 4
**Correctness:** 3
**Technical Novelty And Significance:** 3
**Empirical Novelty And Significance:** 3
**Recommendation:** 8

**Clarity, Quality, Novelty And Reproducibility:**

The paper is very well written and clearly outlines the ideas behind the approach. The test cases are well presented and the various algorithms provided help clarify the method.

**Strength And Weaknesses:**


**Strengths :**

* Combination of intrusive simulation based on a *PINNs-like* scheme and non-linear dimensionality reduction which is a key challenge in reduced order modeling.
* "Mesh free" approach : Although a discretised solution is still necessary for the initialisation of the CROM, the model is able to predict the solution at arbitrary inference points during the simulation.
* The method outperforms classical reduction methods on the proposed test cases

**Weaknesses :**

* The main weakness of the paper are the test cases which might be a bit too simple. 1D diffusion problems are used to showcase the method and the case of the burger's equation clearly presents the interest of using a non-linear dimensionality reduction method. However, the case of the Navier-Stokes equations could be extended to more complex conditions such as the cylinder flow, or the fluidic pinball to demonstrate the performance of the approach on a more realistic test case.
* Tying back to the previous point, issues such as non-markovian effects in the dynamics might appear when dealing with the reduction of complex chaotic problems, which are not treated in this work.
* The name of the method should be changed, as a paper on cluster based reduced order modeling ([1]) already uses the acronym.

*[1] E. Kaiser, B. R. Noack, L. Cordier, A. Spohn, M. Segond, M. Abel, G.Daviller, J. Östh, S. Krajnović and R. K. Niven (2014). Cluster-based reduced-order modelling of a mixing layer. [*Journal of Fluid Mechanics* , **754**](http://journals.cambridge.org/action/displayIssue?jid=FLM&volumeId=754&seriesId=0&issueId=-1), pp 365-414.*

**Summary Of The Paper:**


This work proposes a novel reduced order modeling approach based on deep neural networks. Here, the solution of the problem is approximated by a neural network, rather than a basis of mesh elements as is done in most reduction approaches. Dimensionality reduction is conducted through a *context* vector which is non-linearly encoded from an initial full order solution. Despite using a non linear reduction scheme, the approach manages to use the original model equations to advance the reduced state in time, bypassing the need for the development of a surrogate forecasting system.

**Summary Of The Review:**

The paper proposes a method able to combine non-linear dimensionality reduction and intrusive simulation. Despite the need for more complex test cases, the paper clearly demonstrates the ability of the approach to efficiently model dynamical problems.

---

> ### Author Response · Authors · 2022-11-14
> **Response to Reviewer NfXC**
>
> - Q: More complex test
>
> - A:
>
>   - Yes, we agree that our method can benefit from validation on more challenging caes, such as cylindrical flow and non-markovian behaviors. We have compteded the cylindrical flow experiment and listed the non-markovian effects in the future work section (See Appendix K.5).
>   - As discussed in the general response, our approach faithfully captures the initial steady state, bifurcation, and the vortex shedding regime seen in the cylindrical flow / karman vortex street experiment. See <https://www.dropbox.com/s/bk019kywmx7nw9d/karman_video.mp4?dl=0> and Appendix K.5.
>   - In addition, the ROM community has long been developing techniques addressing chaotic system involving non-markovian effects, such as \[1] adding closure terms to the dynamics, \[2] adopting a Koopman-operator formalism, and \[3] evolving-then-proejct. Our ROM approach is compatible with all of these “fixes” above, since the only difference of between our approach and prior approaches is the network architecture of the manifold parameterization function.
>
> - Q: Paper acronym has been used by other papers.
>
> - A: Great catch! We will change it to “C-ROM” in the final version.
>
> We appreciate reviewer’s suggestion of the cylindrical flow experiment! Please consider raising the rating after taking this experiment into consideration. We look forward to incorporating any suggestions on improving the quality of the manuscript.
>
> \[1] Pan+ Data-driven discovery of closure models, SIAM Journal on Applied Dynamical Systems, 2019
>
> \[2] Lin+ Data-driven model reduction, Wiener projections, and the Koopman-Mori-Zwanzig formalism, Journal of Computational Physics, 2021
>
> \[3] Carlberg+ Efficient non-linear model reduction via a least-squares Petrov--Galerkin projection and compressive tensor approximations, 2011

---

### Author Response · Authors · 2022-11-14
**General Response to All Reviewers**

- We thanks all the reviewers for their thoughtful comments. We would like to take this opportunity to highlight the newly added Karman vortex street experiment, as requested by reviewers (NfXC,YNJM). See <https://www.dropbox.com/s/bk019kywmx7nw9d/karman_video.mp4?dl=0>. More details are listed in Appendix K.5. Karman vortex street is particularly challenging as the flow bifurcates from a steady state to the periodic vortex shedding regime. To capture such an effect, the latent space dynamics needs to be robust under extreme nonlinearity. As reviewer YNJM and \[1] pointed out, many neural-network-based methods, including physics informed neural network (PINN), are well-known to fail in this case. Our approach faithfully captures the initial steady state, bifurcation, and the vortex shedding regime.

- With that, we are glad to know the reviewers find our work to be

  - Novel, interesting, promising, well-motivated, inspiring to many other researchers, and impactful in both the machine learning and the numerical PDE communities (YNJM, yRQN, XNfR, 7uQY)
  - Bridging emerging topics, such as neural-PDE, implicit neural representation, reduced order modeling, physics informed neural network (NfXC, yRQN, XNfR)
  - Outperform previous discretization-dependent ROM approaches in terms of memory, accuracy, the ability to remesh, and applicability to any discretization  (NfXC, YNJM, yRQN)
  - Able to accelerate PDE solutions while enforcing PDE constraints (NfXC, YNJM, yRQN)
  - rich in experiments and fully reproducible (7uQY)
  - Well-written and easy to follow (NfXC, yRQN, XNfR, 7uQY)

- We have addressed the reviewer’s concerns and updated the manuscript (with textual revisions in red) by:

  - discussing the limitations of our work (YNJM)
  - sharing the [source code](https://www.dropbox.com/s/dj8lwk4mp34mk5i/crom_source_code.zip?dl=0) (yRQN)
  - including the training time of our approach as well as the baseline methods (yRQN, 7uQY)
  - discussing how our approach can handling more complex system and demonstrate it experiemntally via the karman vortex street experiment  (NfXC,YNJM)
  - improving the overall clarity via including more details about the latent space dynamics (XNfR)
  - discussing how the training dataset is chosen (yRQN, 7uQY)

We hope the reviewers consider raising their scores after taking these revisions (along with the personalized responses) into consideration.

We also encourage any additional suggestions related to writing or experiments. We truly appreciate it.

\[1] Chuang+, Experience report of physics-informed neural networks in fluid simulations: pitfalls and frustration, arXiv 2022

---

> ### Author Response · Authors · 2022-12-11
> **Update to the General response**
>
> - As the discussion period is coming into an end, we encourage all reviewers to take a look at our previous responses if you haven't already: <https://openreview.net/forum?id=FUORz1tG8Og&noteId=xonm_AciSRQ>
> - Since the previous response, we have also demonstrated advantages over both POD and neural-network-based autoencoders on the Karman vortex street experiment. See <https://www.dropbox.com/s/bk019kywmx7nw9d/karman_video.mp4?dl=0> Overall, our approach faithfully captures the initial steady state, bifurcation, and the vortex shedding regime.

---

### Decision · Program_Chairs · 2023-01-20

**Decision:**

Accept: notable-top-25%

**Justification For Why Not Higher Score:**

The paper offers a nice description of the motivations and of the proposed method. It comes timely to highlight links between surrogate ML models and reduced order models used for PDEs. This is original work which will certainly be of interest to the community, the idea is natural and easy to catch. Why not oral: the paper does not introduce new major  idea or concept.


**Justification For Why Not Lower Score:**

Good paper that deserves acceptation according to all reviewers

**Metareview: Summary, Strengths And Weaknesses:**

The paper introduces a reduced order model for approximating PDEs. The main novelty consists in learning a continuous representation, implemented via an INR (Implicit Neural Representation) method, of the reduced basis instead of a discrete representation in classical, e.g. POD, methods. The form of the PDE is assumed known. For a given PDE, data is generated via a solver and used to train an INR conditioned on a latent state. This INR makes it possible to generate spatially continuous representations at any time. At time t, the PDE is solved for time t+1 using a numerical scheme in the original space on a SMALL sample M of spatial points – this is where time saving occurs w.r.t. high fidelity solvers that require solving on the whole discretized spatial space. The latent space at time t+1 can then be inferred from the solutions at the M points by solving an inverse problem, which allows the INR to generate the full solution at time t+1. The process is then repeated.

All the reviewers have very positive appreciations, highlighting the quality of the exposition, the novelty of the method which combines INR and solvers and the evaluation performed on a variety of equations. The authors added the required precisions and the weak points mentioned by the reviewers during the rebuttal. I propose an accept.



**Note From Pc:**

if the above contains the word "oral" or "spotlight" please see: "oral" presentation means -> notable-top-5% and "spotlight" means -> notable-top-25%. As stated in our emails, we are disassociating presentation type from AC recommendations